# From Linking Homophily and Label Informativeness to Rewiring in GNNs

## Abstract

Message-passing graph neural networks (GNNs) are widely used for node classification. These models learn node representations by aggregating information along the edges of a given graph. A central open question remains which graph properties make message passing effective. While homophily was long viewed as a key ingredient, recent work has increasingly questioned this view, arguing that message passing can remain effective under heterophily when the label distribution is informative, i.e., when a node's label is predictable from its neighbors' labels. In this work, we bridge these perspectives by formally connecting label distribution informativeness and homophily, showing they are *not independent* and characterizing conditions under which strong neighbor-label predictability becomes unlikely at low homophily. Building on this insight, we propose a rewiring framework that increases homophily using a *reference edge set*, providing guarantees on the homophily of the rewired graph and, in regimes we characterize, also provably strengthening neighbor-label predictability. Across diverse heterophilic benchmarks, our approach achieves higher node-classification accuracy than existing rewiring methods in most evaluated settings and remains competitive with specialized heterophily GNNs, while remaining efficient and scalable to large graphs.

## 1 Introduction

Message-passing graph neural networks (GNNs) are a popular tool for node classification tasks: they learn node representations by repeatedly aggregating information from neighbors along the edges of a given graph. Specifically, we consider the transductive setting in which nodes have observed features, edges do not carry features, and the training, validation, and test sets are disjoint subsets of nodes in the same graph. A central and still open question is *which properties of a graph make message passing effective for node classification*, and when message passing should be expected to succeed or fail.

A common earlier explanation centered on *homophily*: the tendency of adjacent nodes to share the same label, which intuitively supports neighborhood aggregation. This view was challenged by the influential ICLR 2022 paper *Is Homophily a Necessity for Graph Neural Networks?* Ma et al. (2022), which argued that homophily is not inherently required: even under low edge homophily, message passing can work when the *label distribution* is favorable, i.e., when a node's label remains predictable from its neighbors' labels. Subsequent work made this notion more explicit by introducing *Label Informativeness* (LI), an information-theoretic proxy for label-distribution informativeness that quantifies how predictive neighbor labels are of a node's label Platonov et al. (2023a), and by re-evaluating heterophily benchmarks and showing that properly tuned standard GNNs can be competitive with specialized heterophily architectures Platonov et al. (2023b). These works have shifted attention from homophily towards informative label distribution as a primary factor for message-passing performance.

However, this shift leaves an important gap: *are homophily and label distribution independent degrees of freedom?* In particular, when should we expect a heterophilic graph to have an informative label distribution under the LI proxy? Without a formal relationship between these quantities, it is unclear whether heterophily

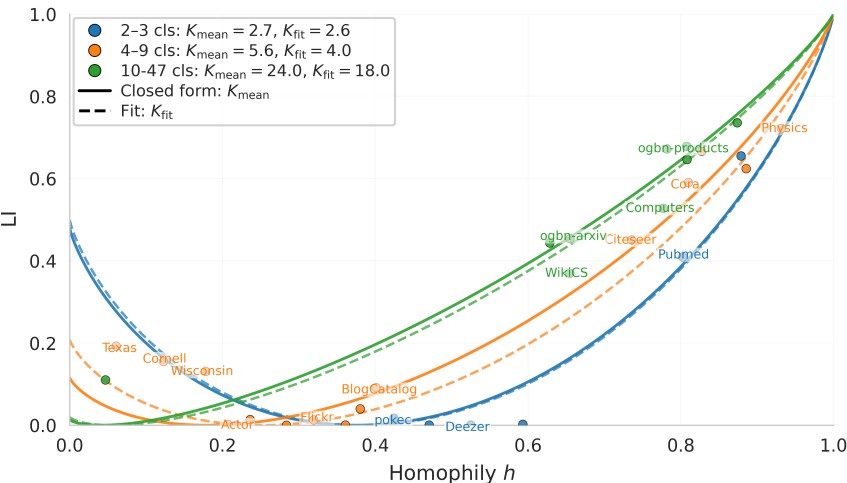

Figure 1: Theoretical and empirical relationship between edge homophily $h$ and label informativeness LI. Points show $(h, \text{LI})$ for 31 standard node-classification benchmarks, grouped by their number of classes $K$, and curves are instances of the closed-form model $\text{LI}(h; K)$ from equation 3, fitted per group together with the curve using the empirical mean $K$. The alignment indicates that the considered benchmarks approximately follow the balanced symmetric model of Theorem 2.2, under which LI at low homophily decreases as the number of classes grows.

with a favorable label distribution is a common regime or an exceptional one, and what this implies for methods that attempt to improve message passing by modifying the graph.

In this work, we attempt to answer these questions by focusing on LI as a standard proxy for label-distribution informativeness, while noting that other proxies could also be defined. We establish a formal connection between homophily and label distribution through LI. We show that both homophily and LI are functionals of the same edge-endpoint label distribution, derive theoretical relationships between them, and show that, under the balanced symmetric model of Theorem 2.2, *high LI is inherently unlikely when homophily is low*. This conclusion is not universal: low homophily and high LI can coexist for general edge-label distributions. Figure 1 shows that the considered benchmarks approximately follow the restricted relationship described by this model.

Beyond linking homophily and LI, we explain why homophily can benefit message passing by relating it to the class separability of node embeddings. We further use controlled simulations to show that, in certain regimes, increasing homophily can improve node classification accuracy even when it degrades LI (i.e., harms the label distribution).

The two observations above motivate improving graph learning *via homophily-increasing rewiring* as an explicit mechanism to enhance message passing. We therefore propose a principled rewiring framework that leverages a *reference edge set* to increase homophily, and we derive guarantees for homophily improvement in the rewired graph. Across a diverse set of heterophilic benchmarks, our rewiring approach achieves higher node-classification accuracy than existing rewiring methods in most evaluated settings and remains competitive with specialized heterophily-oriented GNNs, while remaining efficient and scalable to large graphs.

Graph rewiring remains a debated design choice and is not universally appropriate. Its limitations include the risk of removing task-relevant relations, introducing spurious ones, or altering relations that should be preserved; consequently, it may not improve downstream performance in every setting (Attali et al., 2024).

**Contributions.** Our main contributions are: (i) a formal characterization of the relationship between homophily and label distribution as measured by LI, including a general lower bound and a balanced symmetric model characterizing when LI is limited at low homophily; (ii) a separability-based perspective and simulations clarifying how homophily can benefit message passing beyond LI alone; (iii) a rewiring framework

based on a reference edge set, with provable homophily improvement and strong empirical performance on heterophilic node-classification benchmarks.

## 2 Homophily and Label Distribution

### 2.1 Notation and Definitions

Let $\mathcal{G} = (\mathcal{V}, \mathcal{E})$ denote a graph, where $\mathcal{V}$ is the set of nodes, $\mathcal{E}$ is the set of edges, and $|\mathcal{V}| = n$. For clarity, we present all definitions for the undirected case; the directed case follows by defining the same quantities with respect to directed edge endpoints. Node features are collected in a matrix $\mathbf{X} \in \mathbb{R}^{n \times d}$, where the $i$-th row $x_i \in \mathbb{R}^d$ is the feature vector of node $i$. We focus on node classification with $K$ classes, where the label of node $i$ is $y_i \in \{1, \ldots, K\}$.

Let $\Omega$ denote the set of all allowable node pairs over $\mathcal{V}$. For any edge set $\mathcal{S} \subseteq \Omega$, let $\mathcal{S}^c := \Omega \setminus \mathcal{S}$ denote its complement. Given two edge sets $\mathcal{S}_1, \mathcal{S}_2 \subseteq \Omega$, we use the standard set operations $\mathcal{S}_1 \cup \mathcal{S}_2$, $\mathcal{S}_1 \cap \mathcal{S}_2$, and $\mathcal{S}_1 \setminus \mathcal{S}_2$.

**Edge-endpoint label distribution.** Following Platonov et al. (2023a), we consider the *edge-based* label distribution obtained by sampling an edge $(u, v) \in \mathcal{E}$ uniformly at random and inspecting the labels at its endpoints. Let $C := y_u$ and $C' := y_v$ denote the random variables representing the endpoint labels, and define their joint probability matrix $\mathbf{\Pi} \in [0, 1]^{K \times K}$ by

$$\mathbf{\Pi}_{ij} := \mathbb{P}(C = i, \ C' = j), \qquad i, j \in \{1, \ldots, K\}.$$

For an undirected graph, $\mathbf{\Pi}$ is symmetric: $\mathbf{\Pi}_{ij} = \mathbf{\Pi}_{ji}$.

The corresponding (edge-endpoint) marginals are given by row or column sums:

$$\pi_i := \sum_{j=1}^{K} \mathbf{\Pi}_{ij} \ = \ \sum_{j=1}^{K} \mathbf{\Pi}_{ji}, \qquad i \in \{1, \ldots, K\}.$$

Equivalently, $\pi_i$ is the probability that an endpoint of a uniformly sampled edge has label $i$.

**Edge homophily.** Using this notation, edge homophily is the probability of sampling an edge with a same-class endpoints, i.e.:

$$h(\mathcal{E}) := \sum_{i=1}^{K} \mathbf{\Pi}_{ii}.$$

More generally, for any edge set $\mathcal{S} \subseteq \Omega$, we write

$$h(\mathcal{S}) := \frac{1}{|\mathcal{S}|} \sum_{(u,v) \in \mathcal{S}} \mathbf{1}\{y_u = y_v\}.$$

When the graph is clear from context, we write $h := h(\mathcal{E})$. Unless stated otherwise, reported homophily is computed post hoc over the full edge set using all node labels, including test labels, and is used only for evaluation, not by the rewiring procedure.

**Baseline homophily.** As a baseline, consider the case in which the endpoints are paired at random, so that their respective labels are *independently* determined by the marginals. Let $\boldsymbol{\pi} = (\pi_1, \ldots, \pi_K) \in [0, 1]^K$ denote the endpoint-label marginal distribution. In this random-pairing case, the joint endpoint distribution can be represented by the matrix $\mathbf{\Pi}_0$ defined as

$$\mathbf{\Pi}_0 := \boldsymbol{\pi} \boldsymbol{\pi}^\top, \qquad (\mathbf{\Pi}_0)_{ij} = \pi_i \pi_j.$$

We refer to the resulting homophily, given by

$$h_0 := \sum_{i=1}^{K} (\mathbf{\Pi}_0)_{ii} \ = \ \sum_{i=1}^{K} \pi_i^2$$

as the *baseline homophily.* Intuitively, $h_0$ is the expected homophily obtained by ignoring label correlations across edges and pairing edge endpoints at random according to the marginals ($\pi_i$).

**Label Informativeness (LI).** We use *Label Informativeness* (LI) as a metric that quantifies how predictive neighbor labels are of a node's label Platonov et al. (2023a). Let $\mathcal{H}(C)$ denote the Shannon entropy of the random variable $C$ introduced above, representing the label at a randomly sampled edge endpoint, given by

$$\mathcal{H}(C) := -\sum_{i=1}^{K} \pi_i \log \pi_i.$$

The mutual information between the two random variables of edge endpoints under $\mathbf{\Pi}$ is

$$I(C; C') := \sum_{i=1}^{K} \sum_{j=1}^{K} \mathbf{\Pi}_{ij} \log \frac{\mathbf{\Pi}_{ij}}{\pi_i \pi_j},$$

and LI is the normalized mutual information

$$\text{LI} := \frac{I(C; C')}{\mathcal{H}(C)} \in [0, 1]. \tag{1}$$

Conceptually, LI measures how informative a neighbor's label is about a node's label under $\mathbf{\Pi}$: $\text{LI} = 0$ means neighbor labels carry no information beyond the marginal class distribution, whereas $\text{LI} \approx 1$ indicates that knowing a neighbor's label almost fully determines the node's label.

## 2.2 Linking Homophily and Label Informativeness

Our key observation is that *edge homophily $h$ and label informativeness* LI *are both functionals of the same edge-based joint label distribution* $\mathbf{\Pi}$.

This allows us to relate them through a general lower bound.

**Theorem 2.1** (Lower bound on LI in terms of homophily)**.** *Under the edge-based joint label distribution $\mathbf{\Pi}$ with homophily $h$ and baseline homophily $h_0$ defined above, we have*

$$\text{LI} \geq \frac{2}{\mathcal{H}(C)} \left(h - h_0\right)^2. \tag{2}$$

The proof of this result, along with the proofs of all other results in the paper, is provided in Appendix A.

Theorem 2.1 shows that as soon as the observed homophily $h$ deviates from the baseline $h_0$, either towards more homophily ($h > h_0$) or more heterophily ($h < h_0$), the label informativeness LI must be strictly positive, and the deviation $|h - h_0|$ monotonically raises the quadratic lower bound on LI. Since $h_0 = \sum_i \pi_i^2$ is small when label mass is spread across many classes and becomes large only when one class dominates, it is typically below 0.5 in multi-class graphs with reasonably balanced labels (e.g., $h_0 = 1/K$ in the balanced case). In qualitative terms, in this common regime, heterophilic graphs with low homophily values $h$ are expected to lie relatively close to the small baseline $h_0$, so $|h - h_0|$ is small and the lower bound on LI is weak. Thus, in this common regime where $h_0$ is small, sufficiently high homophily raises the guaranteed minimum LI, whereas at low homophily the bound remains weak and, in the general case, both low and high LI are possible.

**Theorem 2.2** (Closed-form $\text{LI}(h)$ in a balanced $K$-class model)**.** *Consider the following joint distribution $\mathbf{\Pi}$ with homophily $h \in [0, 1]$:*

$$\mathbf{\Pi}_{ii} = \frac{h}{K}, \qquad\qquad i = 1, \dots, K,$$

$$\mathbf{\Pi}_{ij} = \frac{1-h}{K(K-1)}, \qquad\qquad i \neq j,$$

This model induces uniform marginals $\pi_i = \frac{1}{K}$ for each $i \in \{1, \dots, K\}$, hence $\mathcal{H}(C) = \log K$ and $h_0 = \sum_{i=1}^{K} \pi_i^2 = \frac{1}{K}$. Then

$$\mathrm{LI}(h) = \frac{h \log(hK) + (1-h) \log\left(\frac{(1-h)K}{K-1}\right)}{\log K}, \tag{3}$$

where $\log$ is the natural logarithm. In particular, $\mathrm{LI}(h_0) = 0$ and $\mathrm{LI}(1) = 1$.

The balanced $K$-class model of Theorem 2.2 captures a clean, idealized relationship between homophily and label informativeness. The curve $\mathrm{LI}(h)$ is anchored at $\mathrm{LI}(1/K) = 0$ and $\mathrm{LI}(1) = 1$, and is strictly increasing in $h$ on $(1/K, 1)$ for fixed $K$: as homophily grows, neighbors become more informative about labels.

To assess how well this model explains real data, we group 31 well-known benchmark datasets by number of classes, plot empirical $(h, \mathrm{LI})$ pairs, and, for each group, fit the closed-form curve equation 3 with $K$ treated as a free parameter while also plotting the same curve with $K$ fixed to the empirical mean class count of that group (see Fig. 1; dataset details are provided in Appendix B). Across all groups, the fitted $K$ values are comparable to the mean number of classes (i.e., a plausible $K$ can be recovered from $(h, \mathrm{LI})$), and both curves align well with the empirical points. This indicates that the homophily–LI behavior of standard benchmarks lies near the theoretical family $\mathrm{LI}(h; K)$ in equation 3, with low-homophily graphs typically exhibiting low LI. This empirical alignment suggests that, although high-LI/low-homophily graphs are possible in general, they are not typical among the benchmarks considered here, which approximately follow the balanced symmetric relationship of Theorem 2.2.

To illustrate the distinction between the general case and the balanced symmetric model, we construct a controlled four-class stochastic block model (SBM) (Holland et al., 1983). The graph transitions from deterministic heterophilic class pairings, where $h = 0$ and $\mathrm{LI} = 1$, through label-independent random mixing, and then to deterministic same-class connectivity, where $h = 1$ and $\mathrm{LI} = 1$. Figure 2 confirms the high-LI/low-homophily boundary case and illustrates why the conclusion of Theorem 2.2 is restricted to its balanced symmetric assumption. The complete construction is provided in Appendix D.

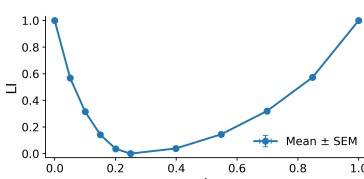

Figure 2: LI versus homophily in the controlled four-class SBM.

Our analysis therefore goes beyond the observation that homophily and LI are both computed from $\mathbf{\Pi}$: Theorem 2.1 provides a general quantitative bound between them, Theorem 2.2 gives an exact closed-form relationship under the balanced model, and Figure 1 shows that this model approximately describes many real-world benchmarks.

## 2.3 Homophily, LI, and GNN Performance

The analysis above links homophily and label informativeness at the level of the edge-label distribution. We now connect these quantities to GNN node classification accuracy. Fig. 3 shows controlled simulations on the heterophilic Cornell graph ($h = 0.12$), where we progressively increase homophily by adding $k$ same-class edges. The edge homophily $h$ increases monotonically with $k$ (left column); as $h$ approaches the baseline $h_0$, both the lower bound in equation 2 and the empirical LI decrease, and as $|h - h_0|$ grows they increase again (middle column), closely tracking the bound. At the same time, GCN node classification accuracy improves monotonically with $h$ (right column), even in the regimes where LI degrades. Thus, these simulations show that increasing homophily improves GNN accuracy, often together with higher LI when $|h - h_0|$ grows, yet accuracy still increases even when LI temporarily degrades as $h$ approaches $h_0$.

To understand why increasing homophily can, in some regimes, benefit message-passing GNNs independently of how LI behaves, we relate homophily to the *smoothness* and *separability* of node embeddings. Let $\mathbf{A}$ be the adjacency matrix, $\mathbf{D}$ the degree matrix, and $\mathcal{L} = \mathbf{D} - \mathbf{A}$ the graph Laplacian. For a node representation matrix $\mathbf{Z} \in \mathbb{R}^{n \times d}$, the standard Dirichlet energy $\mathrm{tr}(\mathbf{Z}^\top \mathcal{L} \mathbf{Z})$ measures how quickly representations vary across edges: low values correspond to smooth, slowly varying signals, while high values reflect rapid changes across the graph.

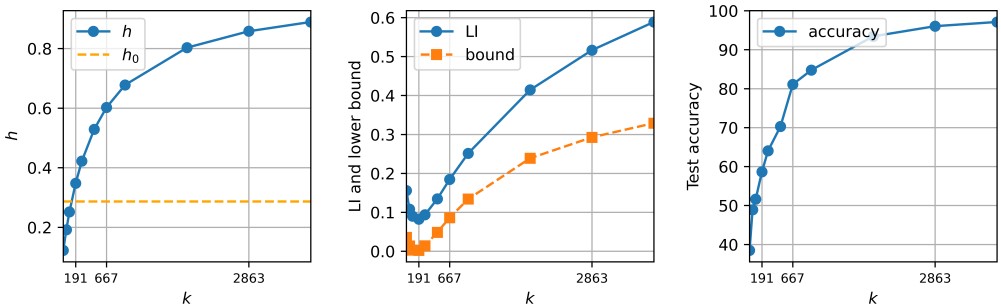

Figure 3: Cornell: homophily, LI, LI lower bound, and GCN test-set node-classification accuracy as a function of the number of added same-class edges $k$. As predicted by Eq. equation 2, LI and its lower bound decrease as $h$ approaches $h_0$ and increase as $h$ moves away from $h_0$; additionally, homophily and GCN test accuracy improve monotonically with $k$, even over regimes where LI degrades.

It is well established that the message passing mechanism in GNNs tends to produce smooth node embeddings ("smoothing is the nature of GNNs" (Chen et al., 2020)). That is, GNNs inherently reduce the smoothness term $\text{tr}(\mathbf{Z}^\top \mathcal{L} \mathbf{Z})$, which is not always beneficial and could lead to over-smoothing. In practice, the quality of node embeddings is often evaluated by their separability, as it reflects how well the nodes can be correctly classified, with linear separability serving as a practical criterion. The following result shows that the ability of GNNs to generate such linearly separable, and therefore effective, node embeddings improves as the homophily of the graph increases.

**Theorem 2.3.** *Let $\mathcal{G}$ be a graph with linearly separable node representations $\mathbf{Z} \in \mathbb{R}^{n \times d}$, and let $\mathbf{W}$ denote the parameters of a linear classifier that separates the classes in the embedding space. Then*

$$\text{tr}(\mathbf{Z}^\top \mathcal{L} \mathbf{Z}) \geq \frac{\alpha_m |\mathcal{E}|}{\|\mathbf{W}\|^2}(1 - h), \tag{4}$$

*where $\alpha_m = \min_{(u,v) \in \mathcal{E}} A_{u,v}$, $\mathbf{A}$ is the adjacency matrix, and $|\mathcal{E}|$ is the number of edges.*

This result explains a benefit of homophily independently of how LI behaves: the right-hand side lower-bounds the Dirichlet energy required for linear separability, and this bound decreases with $h$. Higher homophily therefore makes it easier for message passing to produce embeddings that are both smooth and linearly separable, whereas heterophily raises the risk that smoothing compromises separability. In turn, Theorem 2.3 highlights a key insight: *greater homophily increases the potential of a GNN to learn linearly separable, and hence more effective, node embeddings.*

The LI analysis and the smoothness result therefore address two complementary questions. The former characterizes when homophily constrains the label information carried by graph neighborhoods, while the latter characterizes how homophily affects the compatibility between the smoothing tendency of message passing and class separability.

**Takeaways.** Taken together, the lower bound in Theorem 2.1, the balanced $K$-class model in Theorem 2.2, the empirical trends in Fig. 1, the controlled simulations in Fig. 3, and the smoothness–separability trade-off in Theorem 2.3 show that, for graphs with more than two classes, (i) heterophilic graphs are unlikely to exhibit high label informativeness under the balanced symmetric model, (ii) in some regimes, systematically increasing homophily improves the guaranteed minimum LI, and (iii) empirically, increasing homophily can improve GNN performance even when LI degrades. These observations motivate graph rewiring as a principled way to explicitly enhance homophily (and, in favorable regimes, LI), thereby improving the effectiveness of standard GNNs on heterophilic benchmarks.

## 3 Homophily-Enhancing Rewiring

Among many possible rewiring choices, we present one example that constructs a homophilic *reference edge set* from node features and training labels, assuming the feature space offers a meaningful similarity measure that aligns with the labels, and uses it to rewire the original graph for improved homophily with theoretical guarantees.

### 3.1 Rewiring Framework

We propose a framework to enhance the homophily of a graph $\mathcal{G}$ using a reference edge set $\mathcal{E}_r \subseteq \Omega$. This set contains node pairs that are used as candidates for rewiring. Our approach leverages edge addition and deletion, standard practices in graph rewiring, with the key distinction that these operations are guided by the reference edge set $\mathcal{E}_r$. Under specific conditions on $\mathcal{E}_r$, we demonstrate that this rewiring process guarantees an improvement in the homophily of the resulting rewired edge set $\mathcal{E}^{(k)}$, where $k \in \mathbb{Z}$ indicates the number of added or deleted edges. In Subsection 3.2, we detail how to construct a useful reference edge set from node features and labels.

Given a reference edge set $\mathcal{E}_r$, the rewired graph $\mathcal{G}^{(k)} = (\mathcal{V}, \mathcal{E}^{(k)})$ is obtained by modifying the edge set $\mathcal{E}$ through the addition or deletion of $k$ edges based on $\mathcal{E}_r$. Specifically, $\mathcal{E}^{(k)}$ is defined as:

$$\mathcal{E}^{(k)} = \begin{cases} \mathcal{E} \cup S_k, & \text{if } k > 0, \\ \mathcal{E} \setminus S_{|k|}, & \text{if } k < 0, \end{cases} \tag{5}$$

where $S_k$ is a random subset of $k$ edges from $\mathcal{E}_r \setminus \mathcal{E}$, and $S_{|k|}$ is a random subset of $|k|$ edges from $\mathcal{E} \cap \mathcal{E}_r^c$. Here, $k > 0$ indicates edge addition, and $k < 0$ edge deletion.

**Edge addition.** The following proposition and corollary describe the impact of adding $k$ edges selected at random from $\mathcal{E}_r \setminus \mathcal{E}$ on the homophily of the rewired edge set depending on the homophily of the candidate added edges. For any $k > 0$, let $\mathcal{G}^{(k)}$ be the graph obtained by adding $k$ random edges from $\mathcal{E}_r \setminus \mathcal{E}$ to $\mathcal{G}$, and let $\mathcal{G}^{(k+1)}$ be the graph obtained by adding $k + 1$ such edges. The expected change in homophily stemming from this addition is described in the following result.

**Proposition 3.1.** *If $h(\mathcal{E}_r \setminus \mathcal{E}) > h(\mathcal{E})$, then*

$$\mathbb{E}[h(\mathcal{E}^{(k+1)})] > \mathbb{E}[h(\mathcal{E}^{(k)})] > h(\mathcal{E}).$$

*Otherwise,*

$$\mathbb{E}[h(\mathcal{E}^{(k+1)})] \leq \mathbb{E}[h(\mathcal{E}^{(k)})] \leq h(\mathcal{E}).$$

**Corollary 3.2.** *If $|\mathcal{E}_r| \gg |\mathcal{E}|$, then $h(\mathcal{E}_r) \approx h(\mathcal{E}_r \setminus \mathcal{E})$, and the condition in Prop. 3.1 simplifies to $h(\mathcal{E}_r) > h(\mathcal{E})$.*

**Edge deletion.** The following proposition is similar to Prop. 3.1 but for edge deletion, i.e., where $k < 0$ and the rewired edge set $\mathcal{E}^{(k)}$ is obtained by deleting $|k|$ edges randomly selected from $\mathcal{E} \cap \mathcal{E}_r^c$, where, as defined in Subsection 2.1, $\mathcal{E}_r^c$ denotes the complement of $\mathcal{E}_r$. The expected change in homophily stemming from this deletion is described in the following result.

**Proposition 3.3.** *If $h(\mathcal{E} \cap \mathcal{E}_r^c) < h(\mathcal{E})$, then*

$$\mathbb{E}[h(\mathcal{E}^{(k-1)})] > \mathbb{E}[h(\mathcal{E}^{(k)})] > h(\mathcal{E}).$$

*Otherwise,*

$$\mathbb{E}[h(\mathcal{E}^{(k-1)})] \leq \mathbb{E}[h(\mathcal{E}^{(k)})] \leq h(\mathcal{E}).$$

Thus, when their respective conditions hold, both edge addition and deletion guided by the reference edge set improve homophily in expectation. See Appendix A.6 for concentration bounds and quantitative estimates of the improvement in homophily for both cases.

See Appendix C for controlled simulations on real-world datasets validating these conditions and illustrating how the resulting homophily changes with $k$ (and its downstream effect on GNN accuracy).

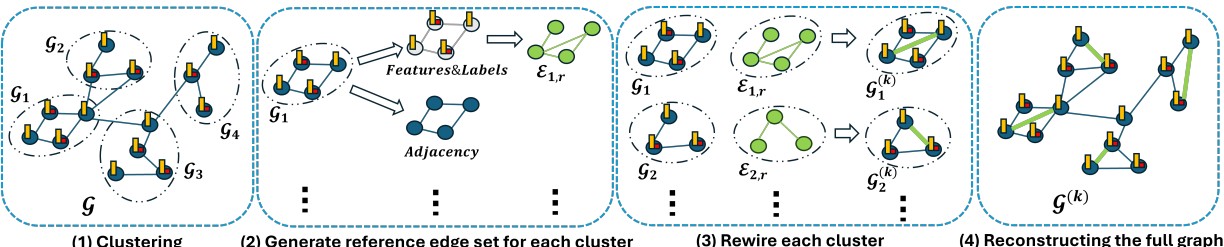

**(1) Clustering** **(2) Generate reference edge set for each cluster** **(3) Rewire each cluster** **(4) Reconstructing the full graph**

Figure 4: Rewiring method overview: (1) Cluster the original graph into clusters. (2) Construct a reference edge set for each cluster using node features (yellow rectangles) and available labels (red squares). (3) Rewire each cluster by modifying edges based on its reference edge set. (4) Reconstruct the fully rewired graph by incorporating inter-cluster edges. We denote the $i$-th cluster as $\mathcal{G}_i$, its reference edge set as $\mathcal{E}_{i,r}$, and the rewired cluster as $\mathcal{G}_i^{(k)}$.

## 3.2 Proposed Method

Given a graph $\mathcal{G} = (\mathcal{V}, \mathcal{E})$, in the transductive node-classification setting considered here, we assume that the nodes $\{1, \ldots, \overline{n}\}$, where $\overline{n} < n$, are labeled (training set), while the nodes $\{\overline{n}+1, \ldots, n\}$ are unlabeled (validation and test sets). Let $\mathbf{X} \in \mathbb{R}^{n \times d}$ denote the node feature matrix, where $x_i \in \mathbb{R}^d$ represents the feature vector of node $i$, and let $\overline{\mathbf{Y}} \in \mathbb{R}^{\overline{n} \times 1}$ denote the labels of the training nodes, where $\overline{y}_i$ is the scalar label of node $i$. The goal is to construct a homophilic reference edge set $\mathcal{E}_r$, which, when applied in the rewiring framework outlined in Subsection 3.1, improves the homophily of the resulting rewired edge set, $h(\mathcal{E}^{(k)})$, in comparison to the original edge set, $h(\mathcal{E})$. If all labels were available, we could construct the ideal same-label reference edge set; however, with access only to training labels $\overline{\mathbf{Y}}$, we aim to construct a reference edge set that is as homophilic as possible. Naïvely building the reference edge set solely based on $\overline{\mathbf{Y}}$ by connecting each pair of training nodes that share the same class label limits generalization and degrades performance when used for rewiring (see Section 4.1), necessitating a more robust approach. Assuming that the feature space offers a meaningful measure of similarity that aligns with the labels, a reference edge set based on feature affinities should be homophilic. Our key idea is to combine node features with the training labels to construct a reference edge set that is not only homophilic but also generalizes to the unlabeled nodes.

To implement this idea, we use the label-driven diffusion approach introduced in Mendelman & Talmon (2025) and propose a diffusion-based method to "complete" the missing labels by propagating label information to the unlabeled nodes through a graph constructed from the fully available node features $\mathbf{X}$. This method captures the shared structure between features and labels, resulting in a reference edge set with higher homophily than the one constructed solely from $\mathbf{X}$ in most cases, as we empirically demonstrate in Appendix H.3. We claim that this diffusion-based construction results in a homophilic reference edge set that is well-suited for our rewiring framework, and we support this claim with extensive empirical results presented in Section 4. While our framework supports any $\mathcal{E}_r$ that satisfies the conditions for improving homophily, the construction proposed here is one possible choice among others.

The first step in constructing the reference edge set $\mathcal{E}_r$ is to build an affinity matrix $\mathbf{W}_D \in \mathbb{R}^{n \times n}$ based on the node feature vectors, where the elements are given by the Gaussian kernel $\mathbf{W}_D(i, j) = \exp\left(-\frac{d^2(x_i, x_j)}{\epsilon}\right)$, for $i, j \in \{1, \ldots, n\}$. Here, $d(\cdot, \cdot)$ is a distance metric in $\mathbb{R}^d$ (e.g., Euclidean distance), and $\epsilon$ is a hyperparameter that controls the scale of the affinity.

Next, we normalize the affinity matrix to obtain the data kernel $\mathbf{D}$, using a standard kernel normalization procedure (Mendelman & Talmon, 2025; Coifman & Lafon, 2006). We compute a diagonal matrix $\mathbf{D}_1$ whose diagonal elements consist of the sum of the rows of $\mathbf{W_D}$ and use it to obtain the intermediate matrix $\widetilde{\mathbf{D}} = \mathbf{D}_1^{-1}\mathbf{W_D}\mathbf{D}_1^{-1}$. Then, we compute another diagonal matrix $\mathbf{D}_2$ consisting of the row sums of $\widetilde{\mathbf{D}}$ and apply a second normalization step, yielding the data kernel $\mathbf{D} = \mathbf{D}_2^{-\frac{1}{2}}\widetilde{\mathbf{D}}\mathbf{D}_2^{-\frac{1}{2}}$. The normalization is written explicitly in Appendix F.1.

---

**Algorithm 1** REFine

---

**Input:** graph $\mathcal{G}$, scale parameter $\epsilon$, cluster size $c$, # added/deleted edges per cluster $k$
Partition $\mathcal{G}$ to $N$ clusters
**for** $i = 1$ **to** $N$ **do**
  1: Construct $\mathbf{\Gamma} = \mathbf{PDP}$ from data and labels
  2: Clip $\mathbf{\Gamma}$ to obtain $\hat{\mathbf{\Gamma}}$ (Eq. 7)
  3: Obtain the reference edge set $\mathcal{E}_{l,r}$ from $\hat{\mathbf{\Gamma}}$
  4: Obtain $\mathcal{E}_l^{(k)}$ using $\mathcal{E}_l$ and $\mathcal{E}_{l,r}$ (Eq. 5)
**end for**
Reconstruct the full rewired graph $\mathcal{G}^{(k)}$

---

For the label-based affinity matrix, since the labels of the validation and test sets are unknown, we define the following binary matrix $\mathbf{W_P} \in \mathbb{R}^{n \times n}$:

$$\mathbf{W_P}(i,j) = \begin{cases} 1, & \text{if } i, j \leq \overline{n} \text{ and } \overline{y}_i = \overline{y}_j, \text{ or } i = j, \\ 0, & \text{otherwise.} \end{cases} \tag{6}$$

Here for simplicity, we assume categorical labels for node classification, but $\mathbf{W_P}$ can be constructed based on continuous labels for graph regression with label affinities. This matrix $\mathbf{W_P}$ is then normalized using the same normalization applied to $\mathbf{W_D}$, yielding the label kernel $\mathbf{P}$.

We consider the following product of the kernels $\mathbf{\Gamma} = \mathbf{PDP}$, representing a label-driven diffusion process with three consecutive steps: propagation within classes using available labels, diffusion via node feature similarity across all nodes, and a final propagation through labels. This process uses node features to "complete" missing labels by propagating label information to unlabeled nodes, capturing the shared geometry between the node features and labels, as demonstrated in Mendelman & Talmon (2025). Here, the three-step product $\mathbf{PDP}$ replaces the original two-factor interpolation, yielding a symmetric kernel suitable for constructing an undirected reference edge set. In addition, the label kernel is binary and assigns zero affinity to differently labeled training nodes, suppressing cross-class relations and favoring homophilic candidate edges. See Appendix F.2, where we visualize the difference between $\mathbf{PDP}$ and $\mathbf{D}$.

Finally, we use $\mathbf{\Gamma}$ to define the reference edge set $\mathcal{E}_r$. Since the elements of $\mathbf{\Gamma}$ are continuous, we clip them to obtain a binary matrix that defines the reference edge set. To this end, we first compute the mean of each row in the matrix $\mathbf{\Gamma}$, given for the $i$-th row by $\mu_i = \frac{1}{n} \sum_{j=1}^{n} \mathbf{\Gamma}(i,j)$. We then clip the elements of each row $i$ based on the mean value $\mu_i$, resulting in the clipped kernel $\hat{\mathbf{\Gamma}}$:

$$\hat{\mathbf{\Gamma}}(i,j) = \begin{cases} 0, & \text{if } \mathbf{\Gamma}(i,j) < \mu_i, \\ 1, & \text{if } \mathbf{\Gamma}(i,j) \geq \mu_i, \end{cases} \tag{7}$$

The binary matrix $\hat{\mathbf{\Gamma}}$ defines the reference edge set $\mathcal{E}_r = \{(i,j) \mid \hat{\mathbf{\Gamma}}(i,j) > 0\}$.

After constructing $\mathcal{E}_r$, it is used to rewire $\mathcal{G}$ by adding or deleting $k$ edges, as described in Section 3.1. REFine is one practical construction within the general reference-edge-set framework, and its homophily guarantee remains conditional on the requirements of Prop. 3.1 and/or Prop. 3.3. Rather than assuming that these conditions always hold, we assess them in practice by estimating $h(\mathcal{E})$ and $h(\mathcal{E}_r)$ using the available training and validation labels; validation labels are used only for this assessment and not to construct the reference edge set. Appendix F.4 shows that these estimates closely approximate the corresponding post-hoc full-label values, while Appendix H.3 reports the homophily of the reference edge sets produced by our construction across the evaluated benchmarks.

Given the obtained reference edge set $\mathcal{E}_r$, we rewire the original graph $\mathcal{G}$ following the framework from Section 3.1. Edge addition ($k > 0$) is performed by randomly selecting $k$ edges from $\mathcal{E}_r \setminus \mathcal{E}$ and adding them to $\mathcal{G}$. Similarly, for edge deletion ($k < 0$), we remove $|k|$ randomly selected edges from $\mathcal{E} \cap \mathcal{E}_r^c$ using the complement edge set $\mathcal{E}_r^c$. The parameter $k$, representing the number of edges added or deleted, is treated as a hyperparameter in our method. This results in the rewired graph $\mathcal{G}^{(k)}$.

**Scaling up.** As our method relies on kernel operations, it is costly on large graphs. To ensure scalability, we cluster the graph $\mathcal{G}$ into $N = \frac{|\mathcal{V}|}{c}$ balanced clusters using the METIS algorithm (Karypis & Kumar, 1998), where $c$ is the cluster size treated as a hyperparameter. This yields the set of graphs $\{\mathcal{G}_l = (\mathcal{V}_l, \mathcal{E}_l)\}_{l=1}^{N}$. Each cluster $\mathcal{G}_l$ is then rewired independently based on its reference edge set $\mathcal{E}_{l,r}$, constructed using the procedure described above. Finally, the full graph is reconstructed by merging all rewired clusters while preserving the original inter-cluster edges.

We term our rewiring algorithm, which uses a reference edge set for refining homophily, *REFine*. Its key steps are summarized in Algorithm 1 and illustrated in Figure 4.

## 4 Experiments

Table 1 compares the node classification performance of our REFine[1] with several well-established rewiring methods: SDRF (Topping et al., 2021), FoSR (Karhadkar et al., 2022), BORF (Nguyen et al., 2023), and DHGR (Bi et al., 2024), across multiple GNN architectures: GCN (Kipf & Welling, 2016), GATv2 (Brody et al., 2021), and APPNP (Gasteiger et al., 2018). We evaluate 11 datasets, ranging from small datasets with hundreds of nodes to large datasets with up to $421k$ nodes, each with varying levels of homophily. The table depicts both the number of nodes and the homophily for each dataset. Our REFine achieves a higher mean node-classification score than the strongest rewiring baseline in most evaluated settings. For the complete table, including the standard error of the mean (SEM), see Appendix H.1. Due to the high computational cost of the competing rewiring methods for large datasets, we adapted all compared methods to use the same clustering strategy as in our approach (Section 3.2). See Appendix H.1 for results on high-homophily datasets (Cora, Citeseer, Pubmed), where our method and the baselines showed no significant improvement over training on the original graph.

To demonstrate the practical benefit of REFine, we compare the performance of standard GNNs combined with REFine to that of specialized GNNs designed for heterophilic graphs. Table 2 presents a comparison of node classification performance on heterophilic graphs ($h < 0.5$). It compares the performance of the top-performing standard GNNs (GCN, GATv2, and APPNP) combined with REFine rewiring (denoted as ST+REFine) for each dataset, against well-established specialized GNNs for heterophilic graphs: MixHop (Abu-El-Haija et al., 2019), GPRGNN (Chien et al., 2020), $H_2$GCN (Zhu et al., 2020), and OrderedGNN (Song et al., 2023). Standard GNNs with REFine achieve the best mean result on four of the eight datasets and remain competitive on the remaining datasets. For the complete table, including the standard error of the mean (SEM), see Appendix H.1.

In Appendix H.2, we compare the homophily of the original edge set $\mathcal{E}$, the reference edge set $\mathcal{E}_r$, and the rewired edge set $\mathcal{E}^{(k)}$ across multiple datasets, demonstrating the effectiveness of our rewiring method in enhancing homophily. See Appendix G for additional implementation details, including parameter choices for our method and the baselines.

### 4.1 Additional Experiments

**Homophily and rewiring effectiveness.** In Appendix J.1, we empirically demonstrate that datasets with lower original homophily tend to show greater test accuracy gains from our rewiring method. This is likely because the reference edge set typically has much higher homophily than the original edge set in such cases, resulting in a significantly more homophilic rewired graph and thus better performance.

**Labels-only ablation.** When the reference edge set is built solely from training labels ($\mathbf{\Gamma} = \mathbf{P}$), $h(\mathcal{E}_r) = 1$, so rewiring necessarily increases homophily. However, this affects only the training subgraph and fails to generalize, leading to worse performance. Appendix J.2 shows that with $\mathbf{\Gamma} = \mathbf{P}$, as $|k|$ increases, homophily improves as expected but test accuracy declines.

**Cluster size.** Larger clusters can yield small performance gains, but with increased runtime. Our experiments use cluster sizes of 100 or 500 depending on graph size. See Appendix J.3 for details.

---

[1]Our code is available in the supplementary materials and will be released on GitHub upon publication.

Table 1: Results on node-classification datasets comparing None (no rewiring), SDRF, FoSR, BORF, DHGR, and REFine. Accuracy is reported for all datasets; for Tolokers and Questions we report ROC AUC due to class imbalance, following Platonov et al. (2023b). "T/O" = timeout; "OOM" = out of memory. Best **bold**; second-best underlined. For REFine, ↑/↓ show the sign of the gain vs. the best baseline. See Appendix H.1 for the complete table with SEM.

| | Cornell | Texas | Wisconsin | Chameleon | Squirrel | BlogCatalog | Actor | BGP | Tolokers | Questions | Genius |
|---|---|---|---|---|---|---|---|---|---|---|---|
| Nodes | 183 | 183 | 251 | 851 | 2223 | 5196 | 7160 | 10k | 11k | 48k | 421k |
| $h$ | 0.12 | 0.06 | 0.17 | 0.23 | 0.2 | 0.4 | 0.21 | 0.28 | 0.59 | 0.84 | 0.59 |
| **GCN** | | | | | | | | | | | |
| None | 51.8 | 59.7 | 57.2 | 41.3 | 40.7 | 77.6 | 28.4 | 53.4 | 77.2 | 65.7 | 83.1 |
| SDRF | 58.4 | 65.4 | 68.6 | 40.6 | **41.5** | 77.9 | 29.2 | 53.9 | 77.6 | OOM | OOM |
| FoSR | 51.6 | 62.4 | 60.5 | 43.1 | 39.7 | 77.4 | 28.1 | 53.3 | 77.4 | 63.3 | 82.2 |
| BORF | 53 | 62.1 | 56.3 | 41.6 | 40.3 | 78 | 28.3 | 52 | 77 | 65.9 | T/O |
| DHGR | 67.8 | 72.7 | 80.6 | 41.1 | 39.1 | 78.3 | **31.4** | 57.3 | 77.2 | 66.9 | OOM |
| REFine (ours) | **71.3** | **79.1** | **82.5** | **44.1** | 41.1 | **85.2** | 31.3 | **59.3** | **78** | **70.3** | **83.8** |
| REFine Gain | ↑3.5 | ↑6.4 | ↑1.9 | ↑1 | ↓0.4 | ↑6.9 | ↓0.1 | ↑2 | ↑0.4 | ↑3.4 | ↑0.7 |
| **GATv2** | | | | | | | | | | | |
| None | 43.7 | 53.2 | 53.3 | 40.8 | 37.4 | 80.4 | 29.6 | 62.3 | 79.3 | 67.4 | 81.7 |
| SDRF | 51 | 61.8 | 63.3 | 39.5 | 37.7 | 83.3 | 29.7 | 63.2 | **79.9** | OOM | OOM |
| FoSR | 46 | 59.7 | 60.9 | 40.1 | 37.7 | 81.6 | 29.2 | 62.8 | 79.5 | **67.6** | 81.2 |
| BORF | 44.6 | 55.1 | 52.5 | 41.2 | 36.7 | 82.2 | 28.6 | 63 | 79.4 | **67.6** | T/O |
| DHGR | **75.1** | 70.2 | 81.7 | 41.8 | 37.6 | 83.2 | 32.8 | 63.2 | 79.2 | OOM | OOM |
| REFine (ours) | 74 | **82.4** | **84.9** | 43.5 | **38.8** | **85.9** | **35.1** | **63.3** | 79.7 | 66.6 | **83.6** |
| REFine Gain | ↓1.1 | ↑12.2 | ↑3.2 | ↑1.7 | ↑1.1 | ↑2.6 | ↑2.3 | ↑0.1 | ↓0.2 | ↓1 | ↑1.9 |
| **APPNP** | | | | | | | | | | | |
| None | 49.4 | 61.9 | 62.1 | 40.2 | 35.4 | 95.7 | 33.8 | 63.6 | 71.1 | 44.1 | 81.9 |
| SDRF | 63.7 | 77 | 75 | 41 | 35.6 | 95.8 | 33.8 | 63.6 | 71.8 | OOM | OOM |
| FoSR | 55.1 | 67 | 68.4 | 41.8 | 35.7 | **95.9** | 33.9 | 63.6 | 71.9 | 44.8 | 81.2 |
| BORF | 55.1 | 65.1 | 66 | 39.6 | 36.2 | 95.5 | 33.6 | 63.4 | 71.4 | 44.5 | T/O |
| DHGR | 70.8 | 74.3 | 81.7 | 43 | 37.9 | 95.4 | 34 | 63.8 | 72.6 | OOM | OOM |
| REFine (ours) | **74.6** | **82.4** | **86** | **44.5** | **38.8** | 95.7 | 34.8 | **64.3** | **73.8** | **47** | **83.6** |
| REFine Gain | ↑3.8 | ↑5.4 | ↑4.3 | ↑1.5 | ↑0.9 | ↓0.2 | ↑0.8 | ↑0.5 | ↑1.2 | ↑2.2 | ↑1.7 |

Table 2: Test accuracy on heterophilic graphs for specialized GNNs vs. ST+REFine. Best is **bold**, second-best is underlined. See Appendix H.1 for the complete table with SEM.

| | Cornell | Texas | Wisconsin | Chameleon | Squirrel | BlogCatalog | Actor | BGP |
|---|---|---|---|---|---|---|---|---|
| MixHop | 71.9 | 79.1 | 83.1 | 43.2 | 39.8 | OOM | **36.2** | 64.3 |
| $H_2$GCN | 73.2 | **82.7** | 82.3 | 41.8 | 40.4 | **96.4** | 30.3 | 64.9 |
| GPRGNN | 70.8 | 81 | 82.5 | 40.9 | 38.5 | 95.7 | 35.4 | **65** |
| OrderedGNN | 70.8 | 77.8 | 82.1 | 38 | 34.3 | 95.7 | 35.8 | **65** |
| ST+REFine (ours) | **74.6** | 82.4 | **86** | **44.5** | **41.1** | 95.7 | 35.1 | 64.3 |

# 5 Complexity and Runtime

REFine has per-cluster complexity $\mathcal{O}(c^3)$, and with $n/c$ clusters this yields $\mathcal{O}(c^2 n)$, where $n$ is the number of nodes and $c$ is the cluster size. Including clustering with METIS (average-case $\mathcal{O}(|\mathcal{E}|)$), the end-to-end complexity is $\mathcal{O}(|\mathcal{E}| + c^2 n)$. On standard sparse benchmarks ($|\mathcal{E}| = \mathcal{O}(n)$), the complexity is $\mathcal{O}(c^2 n)$, and for large graphs with $n \gg c$ the method is effectively linear in $n$. The algorithm parallelizes across clusters, and with $g$ GPUs the parallel implementation runs in $\mathcal{O}((c^2/g) n)$. Appendix I presents complexity and runtime comparisons, demonstrating REFine's significant efficiency gains over existing methods.

# 6 Related Work

**Graph homophily metrics.** Several types of graph homophily metrics have been proposed to capture different aspects of label correlation in graphs. These include edge homophily (Abu-El-Haija et al., 2019),

node homophily (Pei et al., 2020), class homophily (Lim et al., 2021b), neighbor homophily (Gong et al., 2023), and others. Each of these measures highlights a distinct aspect of homophily, depending on the focus of the analysis. In this work, we specifically focus on edge homophily, the simplest and most commonly used measure, which quantifies the fraction of edges that connect nodes with the same label.

**Graph rewiring.** Graph rewiring improves GNN learning by modifying the edge set (Attali et al., 2024), addressing challenges like over-smoothing and over-squashing (Nguyen et al., 2023; Topping et al., 2021). Rewiring can also enhance edge homophily, but few works have explored this, with DHGR (Bi et al., 2024) being the most notable. Most rewiring algorithms rely on predefined structural criteria for edge addition or deletion, rather than learning a new graph structure. In contrast, the DHGR approach involves learning a similarity measure between nodes and solving an optimization problem. While presented as a rewiring method, it aligns more closely with Graph Structure Learning (GSL) (Zhou et al., 2023), which optimizes graph topology. However, our method offers a simpler and more efficient approach to enhancing homophily, avoiding complex optimization while providing theoretical guarantees on the homophily of the rewired graph.

## 7 Conclusion

We establish that edge homophily and label informativeness (LI) are coupled: both are functionals of the same edge-endpoint label distribution, and, under the balanced symmetric model we characterize, strong neighbor-label predictability becomes increasingly unlikely at low homophily as the number of classes grows. Motivated by this connection, we propose REFine, a reference-edge-set-guided rewiring framework with guarantees for homophily improvement and strong empirical gains: across heterophilic benchmarks, standard GNNs combined with REFine outperform prior rewiring methods in most evaluated settings and remain competitive with specialized heterophily GNNs, while remaining scalable.

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

# A    Proofs

## A.1    Proof of Theorem 2.1

*Proof.* Let $\mathbf{\Pi}$ denote the true joint distribution of the endpoint labels $(C, C')$, i.e.,

$$\mathbf{\Pi}_{ij} := \mathbb{P}(C = i, C' = j), \qquad i, j \in \{1, \ldots, K\},$$

with edge-based marginals

$$\pi_i := \mathbb{P}(C = i) = \sum_{j=1}^{K} \mathbf{\Pi}_{ij}, \qquad i \in \{1, \ldots, K\}.$$

For an undirected graph with a uniformly random edge and a uniformly random orientation, $C$ and $C'$ have the same marginal.

Consider the *independence baseline* $\mathbf{\Pi}_0$, where the endpoints are independent but share the same marginals:

$$(\mathbf{\Pi}_0)_{ij} := \mathbb{P}_{\mathbf{\Pi}_0}(C = i, C' = j) = \pi_i \pi_j.$$

Under $\mathbf{\Pi}_0$, the homophily induced purely by the marginals is

$$h_0 := \mathbb{P}_{\mathbf{\Pi}_0}(C = C') = \sum_i \mathbb{P}_{\mathbf{\Pi}_0}(C = i, C' = i) = \sum_i \pi_i^2.$$

Let $A := \{C = C'\}$ denote the event that an edge is homophilous. Then under $\mathbf{\Pi}$ we have

$$\mathbb{P}_{\mathbf{\Pi}}(A) = h := \mathbb{P}(C = C') = \sum_i \mathbf{\Pi}_{ii},$$

and under $\mathbf{\Pi}_0$ we have $\mathbb{P}_{\mathbf{\Pi}_0}(A) = h_0$. Hence

$$\mathbb{P}_{\mathbf{\Pi}}(A) - \mathbb{P}_{\mathbf{\Pi}_0}(A) = h - h_0.$$

The total variation distance between $\mathbf{\Pi}$ and $\mathbf{\Pi}_0$ is

$$\mathrm{TV}(\mathbf{\Pi}, \mathbf{\Pi}_0) := \frac{1}{2} \sum_{i,j} \left| \mathbf{\Pi}_{ij} - (\mathbf{\Pi}_0)_{ij} \right| \geq \left| \mathbb{P}_{\mathbf{\Pi}}(A) - \mathbb{P}_{\mathbf{\Pi}_0}(A) \right| = |h - h_0|.$$

Using Pinsker's inequality with natural logarithms (so that the constant is 2), we obtain

$$D_{\mathrm{KL}}(\mathbf{\Pi} \,\|\, \mathbf{\Pi}_0) \geq 2\,\mathrm{TV}(\mathbf{\Pi}, \mathbf{\Pi}_0)^2 \geq 2\,(h - h_0)^2.$$

By construction, $\mathbf{\Pi}_0$ has product form $(\mathbf{\Pi}_0)_{ij} = \pi_i \pi_j$, so

$$D_{\mathrm{KL}}(\mathbf{\Pi} \,\|\, \mathbf{\Pi}_0) = \sum_{i,j} \mathbf{\Pi}_{ij} \log \frac{\mathbf{\Pi}_{ij}}{\pi_i \pi_j} = I(C; C'),$$

the mutual information between $C$ and $C'$ under the true edge-based joint. Therefore,

$$I(C; C') \geq 2\,(h - h_0)^2.$$

Recalling the definition of label informativeness,

$$\mathrm{LI} = \frac{I(C; C')}{\mathcal{H}(C)}, \qquad \mathcal{H}(C) = -\sum_i \pi_i \log \pi_i,$$

we conclude that

$$\mathrm{LI} = \frac{I(C; C')}{\mathcal{H}(C)} \geq \frac{2}{\mathcal{H}(C)}\,(h - h_0)^2,$$

which is exactly the claimed bound equation 2. $\qquad\square$

## A.2  Proof of Theorem 2.2

*Proof.* Let $(C, C')$ be the endpoint labels of a uniformly sampled edge under the symmetric $K$-class model of Theorem 2.2. By assumption, the joint distribution satisfies

$$\mathbb{P}(C = i, C' = i) = \frac{h}{K}, \qquad\qquad\qquad i = 1, \ldots, K,$$

$$\mathbb{P}(C = i, C' = j) = \frac{1 - h}{K(K - 1)}, \qquad\qquad i \neq j,$$

for some $h \in [0, 1]$.

We first check that this is a valid joint distribution. Summing over all pairs $(i, j)$,

$$\sum_{i=1}^{K} \mathbb{P}(C = i, C' = i) + \sum_{\substack{i,j=1 \\ i \neq j}}^{K} \mathbb{P}(C = i, C' = j) = K \cdot \frac{h}{K} + K(K-1) \cdot \frac{1-h}{K(K-1)} = h + (1-h) = 1.$$

Next, we verify that the marginals are balanced. For any $i \in \{1, \ldots, K\}$,

$$\mathbb{P}(C = i) = \sum_{j=1}^{K} \mathbb{P}(C = i, C' = j)$$

$$= \mathbb{P}(C = i, C' = i) + \sum_{\substack{j=1 \\ j \neq i}}^{K} \mathbb{P}(C = i, C' = j)$$

$$= \frac{h}{K} + (K-1) \cdot \frac{1-h}{K(K-1)}$$

$$= \frac{h}{K} + \frac{1-h}{K}$$

$$= \frac{1}{K}.$$

By symmetry, the same holds for $C'$:

$$\mathbb{P}(C' = j) = \frac{1}{K} \quad \text{for all } j.$$

Hence the edge-end marginals are uniform, and the entropy of $C$ is

$$\mathcal{H}(C) = -\sum_{i=1}^{K} \frac{1}{K} \log\left(\frac{1}{K}\right) = \log K.$$

The homophily under this model is exactly $h$:

$$\mathbb{P}(C = C') = \sum_{i=1}^{K} \mathbb{P}(C = i, C' = i) = K \cdot \frac{h}{K} = h.$$

Let $\mathbf{\Pi}$ denote the joint distribution of $(C, C')$ described above. The independence baseline with the same (uniform) marginals is

$$(\mathbf{\Pi}_0)_{ij} := \mathbb{P}_{\mathbf{\Pi}_0}(C = i, C' = j) = \frac{1}{K^2}, \quad \text{for all } i, j.$$

The mutual information $I(C; C')$ is the Kullback–Leibler divergence between $\mathbf{\Pi}$ and $\mathbf{\Pi}_0$:

$$I(C; C') = D_{\mathrm{KL}}(\mathbf{\Pi} \,\|\, \mathbf{\Pi}_0) = \sum_{i,j=1}^{K} \mathbf{\Pi}_{ij} \log \frac{\mathbf{\Pi}_{ij}}{(\mathbf{\Pi}_0)_{ij}}.$$

We split the sum into diagonal (homophilous) and off-diagonal (heterophilous) terms:

$$I(C; C') = \sum_{i=1}^{K} \mathbf{\Pi}_{ii} \log \frac{\mathbf{\Pi}_{ii}}{(\mathbf{\Pi}_0)_{ii}} + \sum_{\substack{i,j=1 \\ i \neq j}}^{K} \mathbf{\Pi}_{ij} \log \frac{\mathbf{\Pi}_{ij}}{(\mathbf{\Pi}_0)_{ij}}$$

$$= \sum_{i=1}^{K} \frac{h}{K} \log\left(\frac{(h/K)}{1/K^2}\right) + \sum_{\substack{i,j=1 \\ i \neq j}}^{K} \frac{1-h}{K(K-1)} \log\left(\frac{(1-h)/(K(K-1))}{1/K^2}\right).$$

We simplify each part. For the diagonal terms,

$$\frac{h}{K} \log\Big(\frac{h/K}{1/K^2}\Big) = \frac{h}{K} \log(hK),$$

and summing over $i = 1, \ldots, K$ gives

$$\sum_{i=1}^{K} \frac{h}{K} \log(hK) = h \log(hK).$$

For the off-diagonal terms,

$$\frac{1-h}{K(K-1)} \log\Big(\frac{(1-h)/(K(K-1))}{1/K^2}\Big) = \frac{1-h}{K(K-1)} \log\Big(\frac{(1-h)K}{K-1}\Big),$$

and there are $K(K-1)$ pairs $(i,j)$ with $i \neq j$, so

$$\sum_{\substack{i,j=1 \\ i \neq j}}^{K} \frac{1-h}{K(K-1)} \log\Big(\frac{(1-h)K}{K-1}\Big) = (1-h) \log\Big(\frac{(1-h)K}{K-1}\Big).$$

Therefore,

$$I(C; C') = h \log(hK) + (1-h) \log\Big(\frac{(1-h)K}{K-1}\Big).$$

By definition, the label informativeness in this symmetric setting is

$$\mathrm{LI}(h) := \frac{I(C; C')}{\mathcal{H}(C)} = \frac{I(C; C')}{\log K},$$

so substituting the expression above yields

$$\mathrm{LI}(h) = \frac{h \log(hK) + (1-h) \log\Big(\frac{(1-h)K}{K-1}\Big)}{\log K},$$

which is exactly equation 3.

Finally, we check the edge cases. At the independence baseline $h = 1/K$, we have $\mathbf{\Pi}_{ij} = 1/K^2$ for all $i, j$, so $\mathbf{\Pi} = \mathbf{\Pi}_0$ and $I(C; C') = 0$, hence $\mathrm{LI}(1/K) = 0$. At perfect homophily $h = 1$, the joint is supported only on the diagonal with $\mathbb{P}(C = i, C' = i) = 1/K$, so $C' = C$ almost surely and $I(C; C') = \mathcal{H}(C) = \log K$, giving $\mathrm{LI}(1) = 1$. $\qquad\square$

### A.3 Proof of Theorem 2.3

*Proof of Theorem 2.3.* Some of the steps of this proof follow Theorem 3.1 from (Xing et al., 2024), specifically adopting their definition of linearly separable embeddings.

To prove Theorem 2.3, we assume the existence of a linear classifier, parameterized by $\mathbf{W} \in \mathbb{R}^{d \times c}$, where $d$ is the embedding dimension and $c$ is the number of classes, which satisfies the condition $\mathbf{Y} = \mathbf{ZW}$.

Now, we express the term $\sum_{(u,v) \in \mathcal{E}} A_{u,v} \|\mathbf{y}_u - \mathbf{y}_v\|^2$ using the smoothness term:

$$
\begin{aligned}
\sum_{(u,v) \in \mathcal{E}} A_{u,v} \|\mathbf{y}_u - \mathbf{y}_v\|^2 &= \sum_{(u,v) \in \mathcal{E}} A_{u,v} \|\mathbf{z}_u \mathbf{W} - \mathbf{z}_v \mathbf{W}\|^2 && \ldots \mathbf{Y} = \mathbf{ZW} \\
&\leq 2\|\mathbf{W}\|^2 \frac{1}{2} \sum_{(u,v) \in \mathcal{E}} A_{u,v} \|\mathbf{z}_u - \mathbf{z}_v\|^2 \\
&= 2\|\mathbf{W}\|^2 \mathrm{tr}(Z^T \mathcal{L} Z) && \ldots \text{Smoothness definition}
\end{aligned}
$$

where $\mathbf{y}_u$ denotes the one-hot label of node $u$.

Thus we get:

$$
\mathrm{tr}(Z^T \mathcal{L} Z) \geq \frac{1}{2\|\mathbf{W}\|^2} \sum_{(u,v) \in \mathcal{E}} A_{u,v} \|\mathbf{y}_u - \mathbf{y}_v\|^2
$$

Next, defining $\alpha_m = \min_{(u,v) \in \mathcal{E}} A_{u,v}$ as the minimum nonzero entry of $\mathbf{A}$, we obtain:

$$
\begin{aligned}
\mathrm{tr}(Z^T \mathcal{L} Z) &\geq \frac{1}{2\|\mathbf{W}\|^2} \sum_{(u,v) \in \mathcal{E}} A_{u,v} \|\mathbf{y}_u - \mathbf{y}_v\|^2 \\
&\geq \frac{\alpha_m}{2\|\mathbf{W}\|^2} \sum_{(u,v) \in \mathcal{E}} \|\mathbf{y}_u - \mathbf{y}_v\|^2.
\end{aligned}
$$

We now replace $\|\mathbf{y}_u - \mathbf{y}_v\|^2$ with the indicator function $2I(\mathbf{y}_u \neq \mathbf{y}_v)$:

$$
\begin{aligned}
\mathrm{tr}(Z^T \mathcal{L} Z) &\geq \frac{\alpha_m}{2\|\mathbf{W}\|^2} \sum_{(u,v) \in \mathcal{E}} 2I(\mathbf{y}_u \neq \mathbf{y}_v) \\
&= \frac{\alpha_m |\mathcal{E}|}{\|\mathbf{W}\|^2} \left( \frac{1}{|\mathcal{E}|} \sum_{(u,v) \in \mathcal{E}} I(\mathbf{y}_u \neq \mathbf{y}_v) \right) \\
&= \frac{\alpha_m |\mathcal{E}|}{\|\mathbf{W}\|^2} \left( 1 - \frac{1}{|\mathcal{E}|} \sum_{(u,v) \in \mathcal{E}} I(\mathbf{y}_u = \mathbf{y}_v) \right) \\
&= \frac{\alpha_m |\mathcal{E}|}{\|\mathbf{W}\|^2} (1 - h).
\end{aligned}
$$

$\square$

**Relation to label Dirichlet energy.** Let $\mathbf{Y} \in \{0,1\}^{n \times K}$ denote the one-hot label matrix, with $\mathbf{y}_u$ denoting the label vector of node $u$. For an unweighted graph,

$$
\mathrm{tr}(\mathbf{Y}^\top \mathcal{L} \mathbf{Y}) = \frac{1}{2} \sum_{u,v} A_{u,v} \|\mathbf{y}_u - \mathbf{y}_v\|_2^2 = \sum_{u,v} A_{u,v} \mathbf{1}\{y_u \neq y_v\}.
$$

Therefore,
$$\frac{\text{tr}(\mathbf{Y}^\top \mathcal{L}\mathbf{Y})}{\sum_{u,v} A_{u,v}} = 1 - h.$$

Thus, higher homophily corresponds exactly to a smoother label signal over the graph. This also explains the dependence on $1 - h$ in Theorem 2.3: under the condition $\mathbf{Y} = \mathbf{ZW}$, the variation of the labels across edges must be supported by sufficient variation in the node representations.

### A.4 Proof of Proposition 3.1

*Proof of Proposition 3.1.* Let the graph $\mathcal{G}$ have $n + m$ edges, where $n$ edges connect nodes of the same label and $m$ edges connect nodes of different labels.

Let the edge set $\mathcal{E}_r \setminus \mathcal{E}$ have $n' + m'$ edges, where $n'$ edges connect nodes of the same label and $m'$ edges connect nodes of different labels.

Define $x$ as the number of added edges between nodes of the same label:

$$x = \sum_{i=1}^{k} X_i,$$

where $X_i \sim \text{Bernoulli}\left(\frac{n'}{n'+m'}\right)$ are independent and identically distributed (i.i.d.).

The homophily rate of the rewired graph $\mathcal{G}^{(k)}$ is given by:

$$h(\mathcal{E}^{(k)}) = \frac{n + x}{n + m + k}.$$

Taking the expectation over the randomness of $\{X_i\}_{i=1}^{k}$, we have:

$$\mathbb{E}[h(\mathcal{E}^{(k)})] = \frac{n + \mathbb{E}[x]}{n + m + k}. \tag{8}$$

Now calculate $\mathbb{E}[x]$:

$$\mathbb{E}[x] = \mathbb{E}\left[\sum_{i=1}^{k} X_i\right] = \sum_{i=1}^{k} \mathbb{E}[X_i] = k \cdot \frac{n'}{n' + m'}.$$

Substitute $\mathbb{E}[x] = k \cdot \frac{n'}{n'+m'}$ into equation equation 8:

$$\mathbb{E}[h(\mathcal{E}^{(k)})] = \frac{n + k \cdot \frac{n'}{n'+m'}}{n + m + k}.$$

Taking the derivative of this expression with respect to $k$ yields:

$$\frac{\partial \mathbb{E}[h(\mathcal{E}^{(k)})]}{\partial k} = \frac{\frac{n'}{n'+m'} \cdot n + \frac{n'}{n'+m'} \cdot m - n}{(n + m + k)^2}.$$

The derivative is positive when $\frac{n'}{n'+m'} \cdot n + \frac{n'}{n'+m'} \cdot m - n > 0$. Reorganizing this inequality, we find:

$$\frac{n'}{n' + m'} > \frac{n}{n + m}.$$

Thus, when $h(\mathcal{E}_r \setminus \mathcal{E}) > h(\mathcal{E})$, the derivative of $\mathbb{E}[h(\mathcal{E}^{(k)})]$ with respect to $k$ is strictly positive, meaning that increasing the number of added edges increases $\mathbb{E}[h(\mathcal{E}^{(k)})]$. Thus,

$$\mathbb{E}[h(\mathcal{E}^{(k+1)})] > \mathbb{E}[h(\mathcal{E}^{(k)})] > \mathbb{E}[h(\mathcal{E}^{(0)})] = h(\mathcal{E}).$$

Conversely, the derivative is negative when $\frac{n'}{n'+m'} \cdot n + \frac{n'}{n'+m'} \cdot m - n < 0$, which implies:

$$\frac{n'}{n' + m'} < \frac{n}{n + m}.$$

or equivalently $h(\mathcal{E}_r \setminus \mathcal{E}) < h(\mathcal{E})$. In this case, the derivative is strictly negative, so the highest value of $\mathbb{E}[h(\mathcal{E}^{(k)})]$ occurs when $k = 0$. Thus,

$$\mathbb{E}[h(\mathcal{E}^{(k+1)})] < \mathbb{E}[h(\mathcal{E}^{(k)})] < \mathbb{E}[h(\mathcal{E}^{(0)})] = h(\mathcal{E}).$$

$\square$

## A.5 Proof of Proposition 3.3

*Proof of Proposition 3.3.* For simplicity, we define $k > 0$, where $k$ represents the number of deleted edges.

Let the edge set $\mathcal{E} \cap \mathcal{E}_r^c$ have $n^* + m^*$ edges, where $n^*$ edges connect nodes of the same label and $m^*$ edges connect nodes of different labels.

Applying the same procedure as in the proof of Proposition 3.1, we get:

$$\mathbb{E}[h(\mathcal{E}^{(k)})] = \frac{n - k \cdot \frac{n^*}{n^* + m^*}}{n + m - k}.$$

Taking the derivative of this expression with respect to $k$ yields:

$$\frac{\partial \mathbb{E}[h(\mathcal{E}^{(k)})]}{\partial k} = \frac{-\frac{n^*}{n^* + m^*}(n + m) + n}{(n + m - k)^2}.$$

The derivative is positive when $-\frac{n^*}{n^* + m^*}(n + m) + n > 0$. Reorganizing this inequality, we find:

$$\frac{n^*}{n^* + m^*} < \frac{n}{n + m}.$$

This is equivalent to $h(\mathcal{E} \cap \mathcal{E}_r^c) < h(\mathcal{E})$. In this case, the derivative is strictly positive, so increasing the number of deleted edges increases $\mathbb{E}[h(\mathcal{E}^{(k)})]$. Thus, returning to the original notation where $k < 0$ represents deleting $|k|$ edges, we have:

$$\mathbb{E}[h(\mathcal{E}^{(k-1)})] > \mathbb{E}[h(\mathcal{E}^{(k)})] > \mathbb{E}[h(\mathcal{E}^{(0)})] = h(\mathcal{E}).$$

Conversely, the derivative is negative when $-\frac{n^*}{n^* + m^*}(n + m) + n < 0$, which implies:

$$\frac{n^*}{n^* + m^*} > \frac{n}{n + m}.$$

or equivalently $h(\mathcal{E} \cap \mathcal{E}_r^c) > h(\mathcal{E})$. In this case, the derivative is strictly negative, so the highest value of $\mathbb{E}[h(\mathcal{E}^{(k)})]$ occurs when $k = 0$, meaning no edges are deleted. Thus, returning to the original notation where $k < 0$ represents deleting $|k|$ edges, we have:

$$\mathbb{E}[h(\mathcal{E}^{(k-1)})] < \mathbb{E}[h(\mathcal{E}^{(k)})] < \mathbb{E}[h(\mathcal{E}^{(0)})] = h(\mathcal{E}).$$

$\square$

### A.6 Concentration Bounds and Magnitude of Improvement

The concentration bounds follow directly from the proofs of Propositions 3.1 and 3.3 via Hoeffding's inequality.

**Edge addition.** Let $x = \sum_{i=1}^{k} X_i$, where $X_i \sim \text{Bernoulli}\left(\frac{n'}{n'+m'}\right)$ are i.i.d. random variables indicating whether an added edge is homophilic. Then, the homophily of the rewired graph is:

$$h(\mathcal{E}^{(k)}) = \frac{n+x}{n+m+k}$$

Using Hoeffding's inequality:

$$\mathbb{P}\left(|x - \mathbb{E}[x]| \geq \epsilon\right) \leq 2 \exp\left(-\frac{2\epsilon^2}{k}\right)$$

This gives:

$$\mathbb{P}\left(\left|h(\mathcal{E}^{(k)}) - \mathbb{E}[h(\mathcal{E}^{(k)})]\right| \geq \delta\right) \leq 2 \exp\left(-\frac{2\delta^2(n+m+k)^2}{k}\right)$$

and $|\mathcal{E}| = n+m$ so we get:

$$\mathbb{P}\left(\left|h(\mathcal{E}^{(k)}) - \mathbb{E}[h(\mathcal{E}^{(k)})]\right| > \delta\right) \leq 2 \exp\left(-\frac{2\delta^2(|\mathcal{E}|+k)^2}{k}\right)$$

**Edge deletion.** Let $x = \sum_{i=1}^{k} X_i$, where $X_i \sim \text{Bernoulli}\left(\frac{n^*}{n^*+m^*}\right)$ are i.i.d. random variables indicating whether a deleted edge is homophilic. Then, the homophily of the rewired graph is:

$$h(\mathcal{E}^{(k)}) = \frac{n-x}{n+m-k}$$

Using Hoeffding's inequality:

$$\mathbb{P}\left(|x - \mathbb{E}[x]| \geq \epsilon\right) \leq 2 \exp\left(-\frac{2\epsilon^2}{k}\right)$$

This gives:

$$\mathbb{P}\left(\left|h(\mathcal{E}^{(k)}) - \mathbb{E}[h(\mathcal{E}^{(k)})]\right| \geq \delta\right) \leq 2 \exp\left(-\frac{2\delta^2(n+m-k)^2}{k}\right)$$

and $|\mathcal{E}| = n+m$ so we get (defined only for $k < |\mathcal{E}|$):

$$\mathbb{P}\left(\left|h(\mathcal{E}^{(k)}) - \mathbb{E}[h(\mathcal{E}^{(k)})]\right| > \delta\right) \leq 2 \exp\left(-\frac{2\delta^2(|\mathcal{E}|-k)^2}{k}\right)$$

These bounds confirm that for edge addition, when $|\mathcal{E}|^2 \gg k$, the homophily is tightly concentrated around its expectation and the concentration improves as $k$ increases. For edge deletion, when $|\mathcal{E}| \gg k$, the homophily is also tightly concentrated around its expectation, but the concentration becomes looser as $k$ increases.

The magnitude of improvement can also be derived directly from our proofs.

**Edge Addition.** From Proposition 3.1, the expected homophily after adding $k$ edges is:

$$\mathbb{E}[h(\mathcal{E}^{(k)})] = \frac{n + k \cdot \frac{n'}{n'+m'}}{n+m+k}, \quad h(\mathcal{E}) = \frac{n}{n+m}$$

where $n$ and $m$ denote the number of edges in $\mathcal{E}$ connecting nodes of the same and different classes, respectively, and $n'$ and $m'$ denote the corresponding counts in the reference edge set $\mathcal{E}_r$.

Bringing to common denominator and simplifying:

$$\mathbb{E}[h(\mathcal{E}^{(k)})] - h(\mathcal{E}) = \frac{n - h(\mathcal{E})(|\mathcal{E}|+k) + kh(\mathcal{E}_r)}{|\mathcal{E}|+k}$$

by using $n = |\mathcal{E}|h(\mathcal{E})$ we get:

$$\mathbb{E}[h(\mathcal{E}^{(k)})] - h(\mathcal{E}) = \frac{|\mathcal{E}|h(\mathcal{E}) - h(\mathcal{E})(|\mathcal{E}| + k) + kh(\mathcal{E}_r)}{|\mathcal{E}| + k} = \frac{k(h(\mathcal{E}_r) - h(\mathcal{E}))}{|\mathcal{E}| + k}$$

**Edge Deletion.** From Proposition 3.3, the expected homophily after deleting $k$ edges is:

$$\mathbb{E}[h(\mathcal{E}^{(k)})] = \frac{n - k \cdot \frac{n^*}{n^* + m^*}}{n + m - k}, \quad h(\mathcal{E}) = \frac{n}{n + m}$$

where $n^*$ and $m^*$ denote the number of edges in $\mathcal{E} \cap \mathcal{E}_r^c$ connecting nodes of the same and different classes, respectively.

Bringing to common denominator and simplifying:

$$\mathbb{E}[h(\mathcal{E}^{(k)})] - h(\mathcal{E}) = \frac{n - h(\mathcal{E})(|\mathcal{E}| - k) - kh(\mathcal{E} \cap \mathcal{E}_r^c)}{|\mathcal{E}| - k}$$

by using $n = |\mathcal{E}|h(\mathcal{E})$ we get:

$$\mathbb{E}[h(\mathcal{E}^{(k)})] - h(\mathcal{E}) = \frac{k(h(\mathcal{E}) - h(\mathcal{E} \cap \mathcal{E}_r^c))}{|\mathcal{E}| - k}$$

# B   Datasets in Fig. 1

| Dataset | $K$ | $|\mathcal{V}|$ | $|\mathcal{E}|$ | $h$ | $h_0$ | LI |
|---|---|---|---|---|---|---|
| Cora Yang et al. (2016) | 7 | 2 708 | 10 556 | 0.8100 | 0.1698 | 0.5904 |
| Citeseer Yang et al. (2016) | 6 | 3 327 | 9 104 | 0.7355 | 0.1969 | 0.4508 |
| Pubmed Yang et al. (2016) | 3 | 19 717 | 88 648 | 0.8024 | 0.3706 | 0.4093 |
| Roman-empire Platonov et al. (2023b) | 18 | 22 662 | 65 854 | 0.0469 | 0.0895 | 0.1101 |
| Computers Shchur et al. (2018) | 10 | 13 752 | 491 722 | 0.7772 | 0.2987 | 0.5279 |
| Photo Shchur et al. (2018) | 8 | 7 650 | 238 162 | 0.8272 | 0.1964 | 0.6662 |
| CS Shchur et al. (2018) | 15 | 18 333 | 163 788 | 0.8081 | 0.1091 | 0.6467 |
| Physics Shchur et al. (2018) | 5 | 34 493 | 495 924 | 0.9314 | 0.4628 | 0.7222 |
| Amazon-Ratings Platonov et al. (2023b) | 5 | 24 492 | 186 100 | 0.3804 | 0.2793 | 0.0398 |
| penn94 Lim et al. (2021a) | 3 | 41 554 | 2 724 458 | 0.4704 | 0.4604 | 0.0003 |
| Actor Pei et al. (2020) | 5 | 7 600 | 30 019 | 0.2181 | 0.2146 | 0.0002 |
| Flickr Zeng et al. (2019) | 7 | 89 250 | 899 756 | 0.3195 | 0.2488 | 0.0130 |
| DeezerEurope Rozemberczki & Sarkar (2020) | 2 | 28 281 | 185 504 | 0.5251 | 0.5102 | 0.0007 |
| ACM Lv et al. (2021) | 3 | 3 025 | 5 343 | 0.8790 | 0.3372 | 0.6553 |
| Wiki Yang et al. (2020) | 17 | 2 405 | 17 981 | 0.6284 | 0.1040 | 0.4435 |
| BlogCatalog Yang et al. (2020) | 6 | 5 196 | 343 486 | 0.4011 | 0.1768 | 0.0918 |
| WikiCS Mernyei & Cangea (2020) | 10 | 11 701 | 431 726 | 0.6547 | 0.1788 | 0.3697 |
| FacebookPagePage Rozemberczki et al. (2021) | 4 | 22 470 | 342 004 | 0.8854 | 0.3604 | 0.6240 |
| LastFMAsia Rozemberczki & Sarkar (2020) | 18 | 7 624 | 55 612 | 0.8739 | 0.1231 | 0.7359 |
| genius Lim et al. (2021a) | 2 | 421 961 | 984 979 | 0.5932 | 0.6136 | 0.0025 |
| BGP Luckie et al. (2013); Suresh et al. (2021) | 7 | 10 176 | 206 799 | 0.2836 | 0.2846 | 0.0001 |
| pokec Jure (2014) | 3 | 1 632 803 | 30 622 564 | 0.4245 | 0.5013 | 0.0172 |
| Reddit2 Zeng et al. (2019) | 41 | 232 965 | 23 213 838 | 0.7817 | 0.0446 | 0.6721 |
| ogbn-arxiv Hu et al. (2020) | 40 | 169 343 | 1 166 243 | 0.6551 | 0.1604 | 0.4542 |
| ogbn-products Hu et al. (2020) | 47 | 2 449 029 | 61 859 140 | 0.8076 | 0.0789 | 0.6783 |
| DGraphFin Huang et al. (2022) | 4 | 3 700 550 | 4 300 999 | 0.3610 | 0.3496 | 0.0004 |
| cornell Pei et al. (2020) | 5 | 183 | 277 | 0.1227 | 0.2810 | 0.1557 |
| texas Pei et al. (2020) | 5 | 183 | 279 | 0.0609 | 0.2741 | 0.1923 |
| wisconsin Pei et al. (2020) | 5 | 251 | 450 | 0.1778 | 0.2992 | 0.1311 |
| chameleon_filtered Platonov et al. (2023b) | 5 | 890 | 8 854 | 0.2361 | 0.2128 | 0.0139 |
| squirrel_filtered Platonov et al. (2023b) | 5 | 2 223 | 46 998 | 0.2072 | 0.2003 | 0.0013 |

Table 3: Dataset statistics and edge-based measures: edge homophily $h$, independence-baseline homophily $h_0$, and label informativeness (LI).

## C   Validation on Real-World Datasets

**Notation in simulation figures.**   In Figs. 5, 6, 8, 9, 10, and 14, the plot legends use legacy graph-based notation. There, $H(\cdot)$ denotes the same edge-homophily quantity as $h(\cdot)$: $H(\mathcal{G}) = h(\mathcal{E})$, $H(\mathcal{G}_p^{(k)}) = h(\mathcal{E}_p^{(k)})$, and $\mathcal{G}_{r,p}$ denotes the graph $(\mathcal{V}, \mathcal{E}_{r,p})$ induced by the reference edge set. Thus, $H(\mathcal{G}_{r,p}) = h(\mathcal{E}_{r,p})$, and $H(\mathcal{G} \cap \mathcal{G}_{r,p}^c)$ should be read as $h(\mathcal{E} \cap \mathcal{E}_{r,p}^c)$.

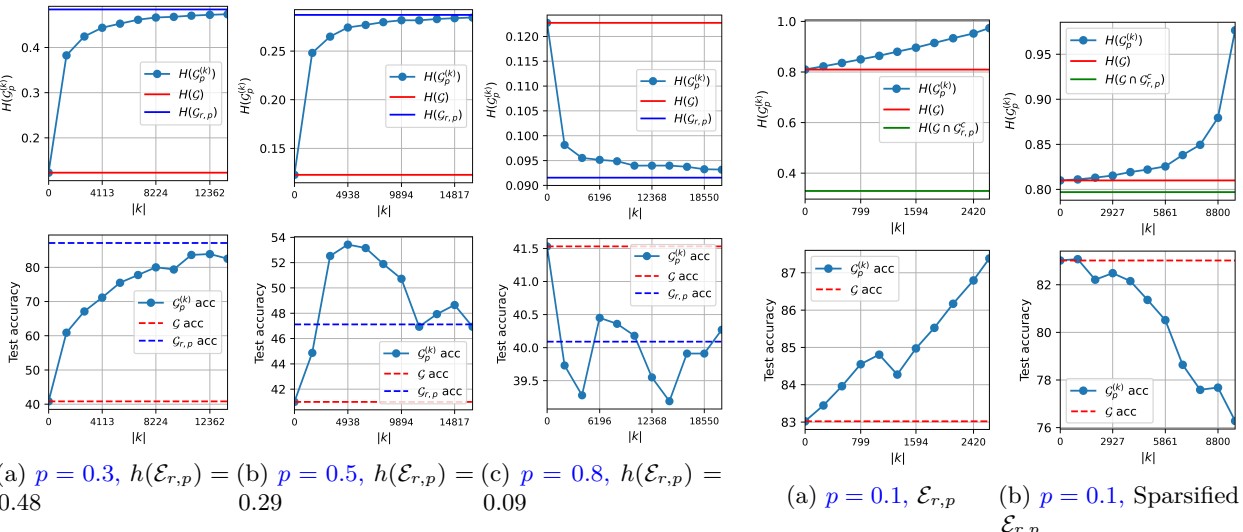

(a) $p = 0.3$, $h(\mathcal{E}_{r,p}) = 0.48$   (b) $p = 0.5$, $h(\mathcal{E}_{r,p}) = 0.29$   (c) $p = 0.8$, $h(\mathcal{E}_{r,p}) = 0.09$

Figure 5: Edge addition on Cornell: First row shows homophily increases with $k$ when the condition holds (blue above red); second row shows its impact on GCN accuracy. $h(\mathcal{E}_{r,p})$ decreases from left to right across the columns.

(a) $p = 0.1$, $\mathcal{E}_{r,p}$   (b) $p = 0.1$, Sparsified $\mathcal{E}_{r,p}$

Figure 6: Edge deletion on Cora: Homophily increases when the condition is met (green line below red).

In this appendix, we provide controlled simulations that empirically demonstrate the behavior predicted in Subsection 3.1.

For validation, we consider several real-world datasets. For each dataset, we consider the original graph $\mathcal{G} = (\mathcal{V}, \mathcal{E})$ obtained from the dataset. We also consider the ideal same-label edge set, where nodes of the same class are connected and nodes of different classes are disconnected. Since we do not have access to all the labels and cannot build such an ideal edge set in practice, we generate reference edge sets, denoted by $\mathcal{E}_{r,p}$, using the following scheme. To control the homophily of the reference edge sets, we use a parameter $p \in (0,1)$ representing the probability of randomly disconnecting or connecting edges relative to the ideal same-label edge set. As $p$ increases, the homophily of the modified reference edge set $\mathcal{E}_{r,p}$ decreases. These reference edge sets are generated only for this controlled validation of the theoretical conditions; they are not constructed by REFine, and $p$ is not a REFine hyperparameter. In addition, we denote by $\mathcal{E}_{r,p}^c$ the complement of $\mathcal{E}_{r,p}$. For each value of $p$, we rewire $\mathcal{G}$ by adding ($k > 0$) or deleting ($k < 0$) $|k|$ edges based on the reference edge set $\mathcal{E}_{r,p}$ following Eq. 5, resulting in the rewired graph $\mathcal{G}_p^{(k)} = (\mathcal{V}, \mathcal{E}_p^{(k)})$. When the condition of Prop. 3.1 is met, or equivalently when $|\mathcal{E}_r| \gg |\mathcal{E}|$ (Cor. 3.2), edge addition is expected to improve homophily. In practice, we use Cor. 3.2, as its assumption holds under the construction of $\mathcal{E}_{r,p}$. Similarly, when the condition of Prop. 3.3 is met, edge deletion is expected to improve homophily. To validate this, we measure and present the obtained empirical homophily of $\mathcal{E}_p^{(k)}$ and evaluate node classification accuracy using GCN, averaging results over 30 runs for each $p$ and $k$.

Figure 5 shows the effect of rewiring with edge addition on the Cornell dataset, where each column represents rewiring using a different $\mathcal{E}_{r,p}$ with a different homophily (controlled by different values of $p$). The first row displays homophily, $h(\mathcal{E}_p^{(k)})$, as a function of $k$, where the red and blue horizontal dashed lines indicate $h(\mathcal{E})$ and $h(\mathcal{E}_{r,p})$, respectively. The second row shows the impact on GCN node classification accuracy ($\mathcal{G}_p^{(k)}$ acc), where the red and blue dashed lines represent the accuracy obtained by $\mathcal{G}$ and using $\mathcal{E}_{r,p}$ as the graph edge set, respectively. In Subfigures 5a and 5b, homophily increases with $k$. This aligns with Cor. 3.2, as the condition

is met – the homophily of the reference edge set indicated by the blue line is larger than the homophily of the original edge set indicated by the red. Conversely, in Subfigure 5c, where the condition is not met (blue line below red), homophily decreases, further supporting the corollary. In the classification accuracy plots, where $h(\mathcal{E}_{r,p})$ is much higher than $h(\mathcal{E})$ (5a), performance improves with increasing $|k|$, but remains lower than training directly using $\mathcal{E}_{r,p}$ as the graph edge set. With $h(\mathcal{E}_{r,p})$ moderately higher than $h(\mathcal{E})$ (5b), accuracy improves up to a point, after which over-smoothing degrades performance. Similar trends are observed in other datasets (see Appendix E). When the condition is not met (5c), edge addition reduces both homophily and performance.

Figure 6 shows the effect of edge deletion on the Cora dataset. The first row displays $h(\mathcal{E}_p^{(k)})$ as a function of $k$, where the red and green horizontal lines represent $h(\mathcal{E})$ and $h(\mathcal{E} \cap \mathcal{E}_{r,p}^c)$, respectively. The second row shows the impact on GCN node classification accuracy. We observe that $h(\mathcal{E}_p^{(k)})$ aligns with Prop. 3.3: removing edges increases homophily when the condition is met (green line below red), and decreases it otherwise. However, higher homophily does not always improve GCN performance, as over-squashing can occur. In one scenario (6a), edge deletion based on $\mathcal{E}_{r,p}$ with $p = 0.1$ improves performance. In the second scenario (6b), we see that a sparse version of the same $\mathcal{E}_{r,p}$ (with 90% edges removed) leads to performance degradation due to the excessive removal of same-class edges, raising the risk of over-squashing (despite the improved homophily).

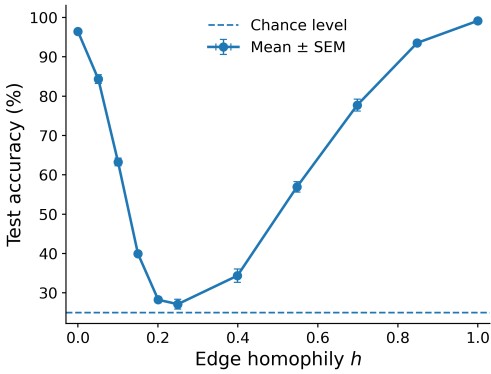
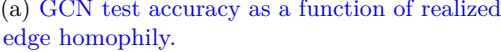

(a) GCN test accuracy as a function of realized edge homophily.

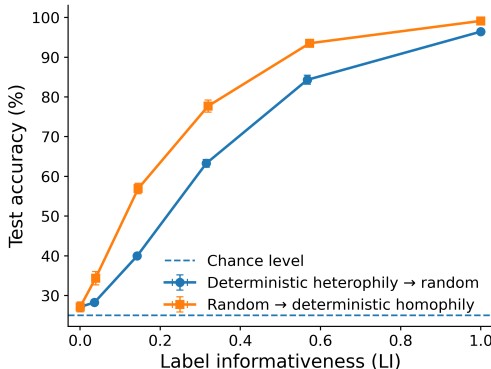

(b) GCN test accuracy as a function of LI along the two mixing branches.

Figure 7: GCN performance along the controlled SBM transition. The dashed line denotes four-class chance accuracy. Results show mean ± SEM over five random seeds.

## D   Synthetic High-LI/Low-Homophily Experiment

The purpose of this experiment is to demonstrate a boundary case in which very low homophily can coexist with high label informativeness. In particular, we construct a graph with $h = 0$ and $\mathrm{LI} = 1$, showing that low homophily does not universally imply an uninformative neighborhood-label distribution. We then continuously interpolate from this deterministic heterophilic regime through label-independent mixing to deterministic homophily.

We use a balanced stochastic block model with four classes and 250 nodes per class. We define three class-level edge-probability matrices: $B_{\mathrm{het}}$, in which classes are connected only through the deterministic pairs $1 \leftrightarrow 2$ and $3 \leftrightarrow 4$; $B_{\mathrm{rand}}$, in which neighbor labels are independent of source labels in expectation; and $B_{\mathrm{hom}}$, in which edges occur only within classes. All three matrices are normalized to yield expected degree 12.

We interpolate between these matrices according to

$$B(t) = \begin{cases} (1 - 2t)B_{\mathrm{het}} + 2tB_{\mathrm{rand}}, & 0 \leq t \leq \frac{1}{2}, \\ (2 - 2t)B_{\mathrm{rand}} + (2t - 1)B_{\mathrm{hom}}, & \frac{1}{2} < t \leq 1, \end{cases}$$

and evaluate $t \in \{0, 0.1, \ldots, 1\}$. At $t = 0$, every class connects to exactly one different class, so the graph has $h = 0$ while the neighbor label fully determines the source label, yielding $\mathrm{LI} = 1$. At $t = 0.5$, endpoint labels are independent in expectation, whereas at $t = 1$, all edges are within classes and therefore $h = 1$ and $\mathrm{LI} = 1$. The resulting relationship between homophily and LI is shown in Fig. 2.

Node features are 16-dimensional class-dependent means with additive Gaussian noise of standard deviation 2. For each class, 20 nodes are used for training, 30 for validation, and the remaining nodes for testing. We train a two-layer GCN with 32 hidden units using Adam, with learning rate 0.01, weight decay $5 \times 10^{-4}$, and early stopping. The expected degree, features, data splits, and model initialization are controlled across $t$ within each run. Results report mean ± SEM over five random seeds.

Figure 7a shows an approximately U-shaped accuracy curve: performance is high at both deterministic endpoints and approaches chance near label-independent random mixing. Figure 7b further shows that comparable LI values can correspond to different GCN accuracies: accuracy is higher on the branch from random mixing toward deterministic homophily than on the branch from deterministic heterophily toward random mixing. Thus, the experiment first establishes the central boundary case that $h = 0$ can coexist with $\mathrm{LI} = 1$, and additionally shows that LI alone does not fully determine the behavior of a standard message-passing GNN. This complements the Cornell experiment in Fig. 3, where accuracy improves with homophily even over a regime in which LI decreases.

## E  Additional Simulations

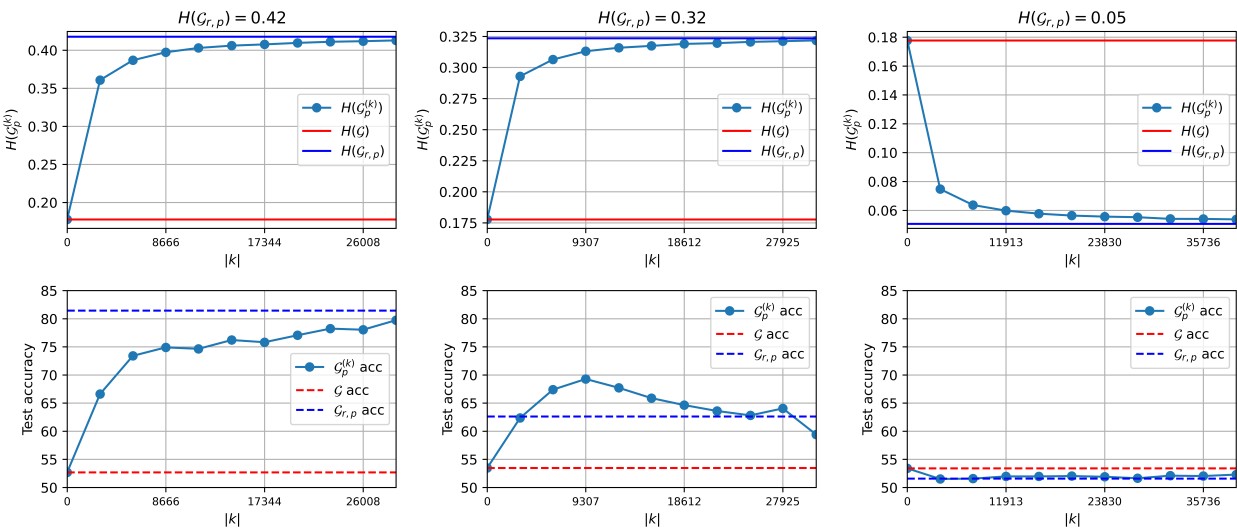

Figure 8: Simulation of edge addition on the Wisconsin dataset. The columns use $p \in \{0.4, 0.5, 0.9\}$ from left to right.

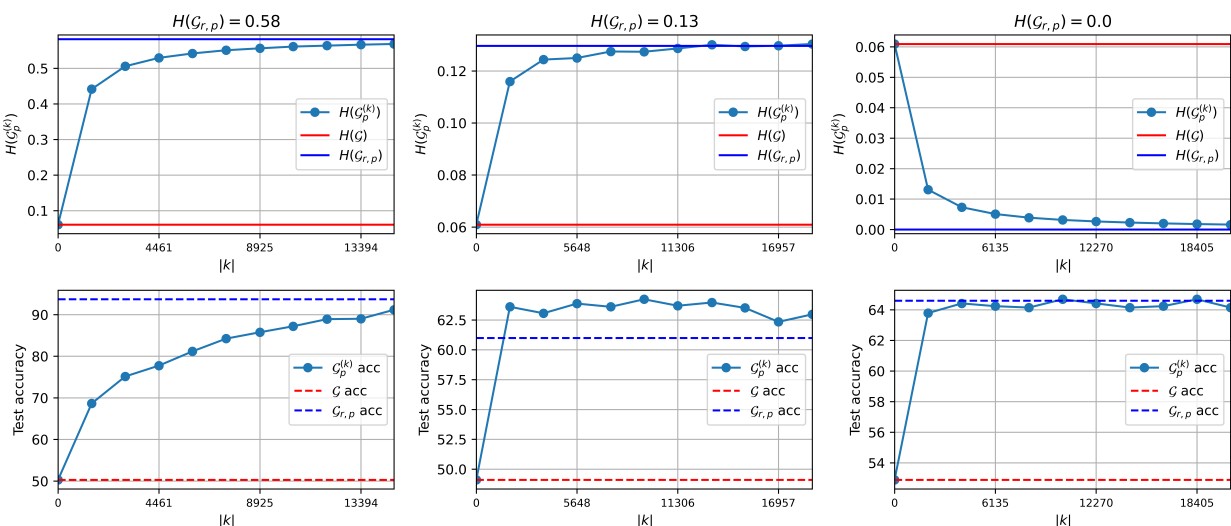

Figure 9: Simulation of edge addition on the Texas dataset. The columns use $p \in \{0.3, 0.8, 1.0\}$ from left to right.

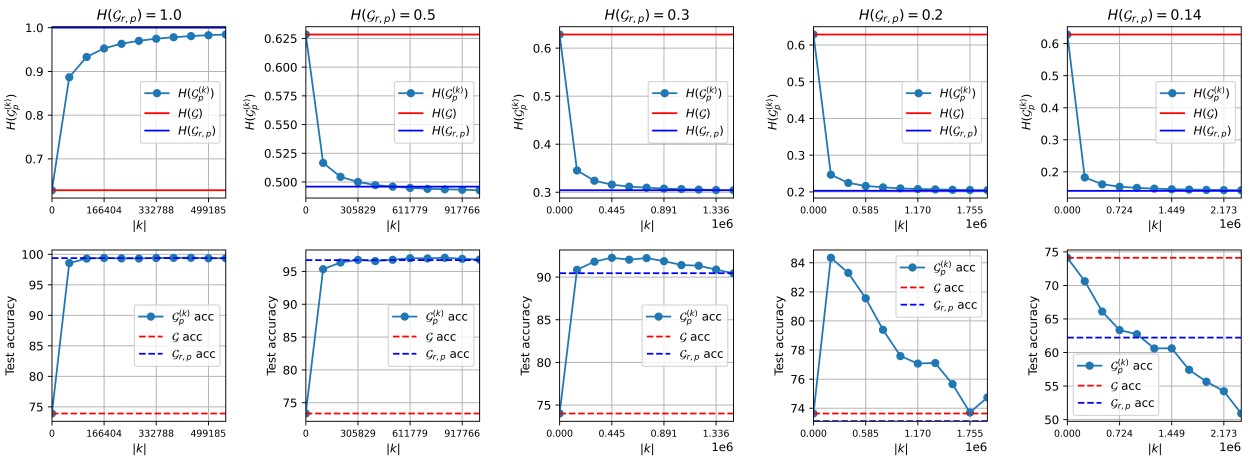

Figure 10: Simulation of edge addition on the Wiki dataset. The columns use $p \in \{0.0, 0.1, 0.2, 0.3, 0.4\}$ from left to right.

# F Additional Method Details

## F.1 Explicit Kernel Normalization

For completeness, the normalization of the data affinity matrix $\mathbf{W}_D$ is given explicitly by

$$(\mathbf{D}_1)_{ii} := \sum_{j=1}^{n} \mathbf{W}_D(i,j), \qquad \widetilde{\mathbf{D}} := \mathbf{D}_1^{-1} \mathbf{W}_D \mathbf{D}_1^{-1},$$

$$(\mathbf{D}_2)_{ii} := \sum_{j=1}^{n} \widetilde{\mathbf{D}}(i,j), \qquad \mathbf{D} := \mathbf{D}_2^{-1/2} \widetilde{\mathbf{D}} \mathbf{D}_2^{-1/2}.$$

The label kernel $\mathbf{P}$ is obtained by applying the same normalization to $\mathbf{W}_P$.

## F.2 Visualization of Kernel Differences

## F.3 Differences Between Our Graph Construction and Label-Driven Diffusion

Our graph construction method using feature vectors and training labels differs from label-driven diffusion (Mendelman & Talmon, 2025) in two key ways: first, we apply a three-step diffusion process to ensure symmetry; second, we set inter-class distances to infinity in the label affinity kernel to promote intra-class connections.

## F.4 Approximating Homophily Using a Sampled Graph

Throughout the paper, the "true" homophily reported in figures and tables refers to the full-graph quantity computed using all ground-truth labels, including test labels, and is used only for post hoc evaluation. In contrast, the sampled construction below estimates this quantity using training and validation labels, without access to test labels.

To check whether the conditions for enhancing homophily hold, we can approximate $h(\mathcal{E})$ and $h(\mathcal{E}_r)$ using the available training and validation labels (the validation labels are used solely for verifying homophily, not as part of the method). We construct a sampled graph $\mathcal{G}^s = (\mathcal{V}^s, \mathcal{E}^s)$ and a sampled reference edge set $\mathcal{E}_r^s$, using validation set nodes whose labels are withheld when constructing $\mathbf{P}$ but are known for evaluation. We randomly sample training nodes such that the ratio of unlabeled validation nodes to sampled labeled nodes matches the ratio of unlabeled (validation and test) nodes to labeled nodes in the full graph. The edge set $\mathcal{E}^s$ consists

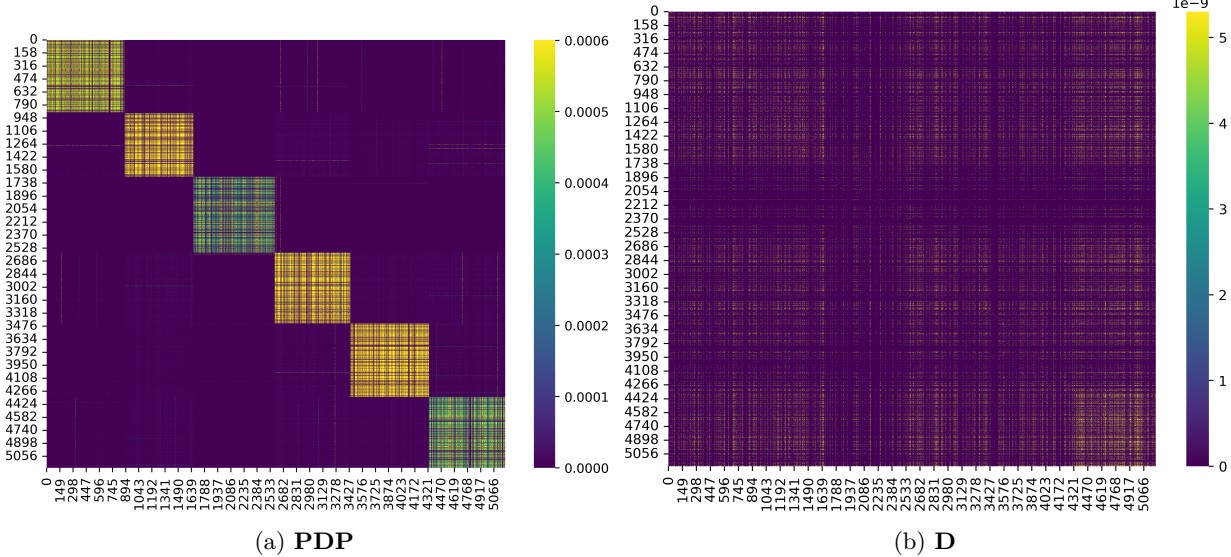

(a) **PDP**  (b) **D**

Figure 11: Heatmap visualizations of the kernels **PDP** and **D** for the BlogCatalog dataset. The rows and columns are sorted by label, with an ideal heatmap showing high-value diagonal blocks for each class. The **PDP** heatmap exhibits better class separation compared to **D**, reflecting improved structure.

of edges from $\mathcal{E}$ connecting pairs of nodes in $\mathcal{V}^s$, and the sampled reference edge set is $\mathcal{E}_r^s = \mathcal{E}_r \cap (\mathcal{V}^s \times \mathcal{V}^s)$. Since the probability of an edge connecting nodes of the same label should remain consistent between the full edge set and its sampled version, $h(\mathcal{E}^s)$ approximates $h(\mathcal{E})$ and $h(\mathcal{E}_r^s)$ approximates $h(\mathcal{E}_r)$. Thus, $\mathcal{G}^s$ and $\mathcal{E}_r^s$ allow us to estimate the homophily quantities used to verify the assumptions underlying our rewiring framework; in particular, they allow us to check the simplified edge-addition condition in Corollary 3.2, and analogous sampled quantities can be used for the edge-deletion condition in Proposition 3.3.

In Table 4, we report the approximated values on real-world datasets and demonstrate that the approximation closely reflects the true homophily. This validates the suitability of the sampled graph and sampled reference edge set for analyzing and verifying the assumptions underlying our rewiring framework.

Table 4: Comparison of true and sampled homophily values. The sampled graph $\mathcal{G}^s$ and the sampled reference edge set $\mathcal{E}_r^s$ provide a close approximation to $h(\mathcal{E})$ and $h(\mathcal{E}_r)$, respectively.

|  | Cornell | Texas | Wisconsin | BlogCatalog |
|---|---|---|---|---|
| $h(\mathcal{E})$ | 0.12 | 0.06 | 0.17 | 0.4 |
| $h(\mathcal{E}^s)$ | 0.14 | 0.08 | 0.2 | 0.4 |
| $h(\mathcal{E}_r)$ | 0.4 | 0.49 | 0.51 | 0.23 |
| $h(\mathcal{E}_r^s)$ | 0.41 | 0.48 | 0.5 | 0.23 |

## G  Experiment Settings

All evaluated datasets are available in PyTorch-Geometric. When official data splits with at least five splits were provided, we used the official splits from PyTorch-Geometric. For datasets without official splits or with fewer than five, we randomly partitioned them into five train-validation-test splits (60%-20%-20%). For Chameleon and Squirrel, we used the filtered versions and corresponding splits provided by Platonov et al. (2023b), as the original versions suffer from train-test data leakage.

For our REFine and all rewiring baselines, we apply our clustering strategy where, for graphs with fewer than 1,000 nodes, we set $c = |\mathcal{V}|$ (no clustering), for graphs with 1,000 to 25,000 nodes, we set $c = 500$, and for graphs with more than 25,000 nodes, we set $c = 100$.

We perform a grid search to optimize hyperparameters on the validation set and report test accuracy with standard error of the mean (SEM). The hidden dimension is set to 32, the learning rate is selected from $\{10^{-4}, 10^{-3}, 10^{-2}, 10^{-1}\}$, and weight decay is searched within $\{10^{-4}, 10^{-3}, 10^{-2}\}$. All models are trained using the Adam optimizer.

For all architectures, we used ReLU activation. Specifically, GCN consists of 2 GCN layers, GATv2 comprises 2 GATv2 layers, and APPNP includes 2 linear layers followed by an APPNP propagation layer with $K = 10$ and $\alpha = 0.1$.

MixHop consists of two MixHopConv layers, each using the default powers $[0, 1, 2]$, followed by a linear output layer. To accommodate the concatenation of three feature sets per layer, the hidden dimension is divided by 3. $H_2$GCN includes a linear feature embedding layer followed by two $H_2$GCNConv layers. Since each $H_2$GCNConv layer concatenates two embeddings, the hidden dimension is divided by 2. The final output layer is a linear projection applied to the concatenated features across all layers. GPRGNN consists of a 2-layer MLP followed by a GPR propagation module with the default $K = 10$, $\alpha = 0.1$, and initialization set to Random. OrderedGNN consists of an input linear transformation layer followed by two OrderedConv layers, each using a temporal matching module composed of a linear layer and layer normalization. The hidden dimension is divided into four chunks to enable chunk-wise updates. A final linear output layer is applied after the OrderedConv blocks.

We transform all directed datasets into undirected ones before applying rewiring and training, as METIS operates only on undirected graphs.

All experiments were conducted using Python on NVIDIA DGX A100 systems, each equipped with A100 GPUs and 512 GB of RAM, with T/0 reached after 24 hours of execution.

**REFine hyperparameters.** For the scale parameter $\epsilon$, we search for the optimal value in $\{1e - 8, 1e - 7, 1e - 6, 1e - 5, 1e - 4, 1e - 3, 1e - 2, 1e0, 1e + 1, 1e + 2\}$. For each dataset, we select $\epsilon$ on the validation set using a fixed $k$ at the center of the considered $k$ range, and then keep the selected $\epsilon$ fixed while tuning $k$. Thus, $\epsilon$ is selected once per dataset and is not re-selected separately for each value of $k$. The selected values are reported in Table 5. We select the best option between using both data and training labels ($\mathbf{\Gamma} = \mathbf{PDP}$) and using only the data ($\mathbf{\Gamma} = \mathbf{D}$). The label kernel $\mathbf{P}$ is constructed using only the training labels from each data split. Additionally, we choose between edge addition and edge deletion.

For $|k|$, the number of added or deleted edges, we search for the optimal value in $\{0.1m, 0.3m, 0.5m, 0.7m, 0.9m, m\}$, where $m = |\mathcal{E}_r|$ (the number of edges in the reference edge set) when adding edges, or $m = |\mathcal{E} \cap \mathcal{E}_r^c|$ (the number of common edges between the original edge set and the complement of the reference edge set) when deleting edges.

Since $\epsilon$ is the bandwidth of a Gaussian kernel, it can also be initialized using the standard median heuristic,

$$\epsilon_{\text{med}} = \text{median}_{i<j} \, d^2(x_i, x_j).$$

For datasets with more than 1,000 nodes, we estimate this quantity from 1,000 uniformly sampled nodes. As shown in Table 5, the selected $\epsilon$ and $\epsilon_{\text{med}}$ lie in the same or adjacent orders of magnitude for 10 of the 11 datasets, with Genius as the exception. Thus, the median heuristic can generally be used to restrict the search to a few nearby scales rather than the full numerical range.

Table 5: Dataset-specific values of $\epsilon$ used in the main experiments and the Gaussian-kernel median heuristic $\epsilon_{\mathrm{med}}$. For datasets with more than 1,000 nodes, $\epsilon_{\mathrm{med}}$ is estimated from 1,000 uniformly sampled nodes.

| Dataset | $\epsilon$ used | $\epsilon_{\mathrm{med}}$ |
|---|---|---|
| Cornell | $10^2$ | $1.28 \times 10^2$ |
| Texas | $10^2$ | $1.04 \times 10^2$ |
| Wisconsin | $10^2$ | $1.25 \times 10^2$ |
| Chameleon | $10^2$ | $1.70 \times 10^1$ |
| Squirrel | $10^1$ | $2.40 \times 10^1$ |
| BlogCatalog | $10^1$ | $1.11 \times 10^2$ |
| Actor | $10^0$ | $8.00 \times 10^0$ |
| BGP | $10^1$ | $1.09 \times 10^1$ |
| Tolokers | $10^{-2}$ | $1.31 \times 10^{-1}$ |
| Questions | $10^{-5}$ | $3.91 \times 10^{-6}$ |
| Genius | $10^{-8}$ | $2.39 \times 10^{-4}$ |

**SDRF hyperparameters.** For each hyperparameter, we search within the range reported in the original paper. Specifically, for the removal bound $(C^+)$, we search for the optimal value in $\{0.5, 1, 10, 20, 40\}$; for the $\tau$ parameter, in $\{50, 100, 200\}$; and for the maximum number of iterations, we define $n$ as the number of nodes in each dataset or the cluster size when using our clustering adaptation for large datasets, and search for the optimal value in $\{0.1n, 0.3n, 0.5n, 0.7n, 0.9n, n\}$.

**FoSR hyperparameters.** We search for the optimal value of the maximum number of iterations in $\{50, 100, 150, 200, 300\}$, based on the range reported in the original paper.

**BORF hyperparameters.** For each hyperparameter, we search within the range reported in the original paper. Specifically, for the number of added edges $(h)$, we search for the optimal value in $\{20, 30\}$; for the number of deleted edges $(k)$, in $\{10, 20, 30\}$; and for the number of batches $(n)$, in $\{2, 3\}$.

# H    Additional Results and Full Tables

## H.1    Complete results

Table 6: Complete results with standard error of the mean (SEM) for datasets containing up to 5,500 nodes. "T/O" indicates a timeout and "OOM" indicates out of memory. Best results are bolded.

|  | Cornell | Texas | Wisconsin | Chameleon | Squirrel | BlogCatalog |
|---|---|---|---|---|---|---|
| Nodes | 183 | 183 | 251 | 851 | 2223 | 5196 |
| $h$ | 0.12 | 0.06 | 0.17 | 0.23 | 0.2 | 0.4 |
| GCN | | | | | | |
| None | $51.8 \pm 1.3$ | $59.7 \pm 2.6$ | $57.2 \pm 1.9$ | $41.3 \pm 0.6$ | $40.7 \pm 0.4$ | $77.6 \pm 0.8$ |
| SDRF | $58.4 \pm 2.2$ | $65.4 \pm 1.8$ | $68.6 \pm 0.8$ | $40.6 \pm 1.0$ | $\mathbf{41.5 \pm 0.6}$ | $77.9 \pm 0.7$ |
| FoSR | $51.6 \pm 2.5$ | $62.4 \pm 1.9$ | $60.5 \pm 1.3$ | $43.1 \pm 1.1$ | $39.7 \pm 0.6$ | $77.4 \pm 0.6$ |
| BORF | $53 \pm 2.7$ | $62.1 \pm 1.8$ | $56.3 \pm 2.3$ | $41.6 \pm 1$ | $40.3 \pm 0.6$ | $78 \pm 0.5$ |
| DHGR | $67.8 \pm 2.1$ | $72.7 \pm 1.9$ | $80.6 \pm 1.8$ | $41.1 \pm 1.1$ | $39.1 \pm 0.3$ | $78.3 \pm 0.5$ |
| REFine | $\mathbf{71.3 \pm 1.5}$ | $\mathbf{79.1 \pm 1.6}$ | $\mathbf{82.5 \pm 1.6}$ | $\mathbf{44.1 \pm 1.1}$ | $41.1 \pm 0.7$ | $\mathbf{85.2 \pm 0.3}$ |
| GATv2 | | | | | | |
| None | $43.7 \pm 2.8$ | $53.2 \pm 3.2$ | $53.3 \pm 1.8$ | $40.8 \pm 0.8$ | $37.4 \pm 0.6$ | $80.3 \pm 0.6$ |
| SDRF | $51 \pm 2.4$ | $61.8 \pm 1.3$ | $63.3 \pm 1.3$ | $39.5 \pm 1.4$ | $37.7 \pm 0.6$ | $83.3 \pm 0.9$ |
| FoSR | $46 \pm 2.2$ | $59.7 \pm 1.6$ | $60.9 \pm 1.8$ | $40.1 \pm 0.6$ | $37.7 \pm 0.4$ | $81.6 \pm 1.4$ |
| BORF | $44.6 \pm 2.3$ | $55.1 \pm 2.5$ | $52.5 \pm 2.1$ | $41.2 \pm 1.2$ | $36.7 \pm 0.6$ | $82.2 \pm 1.4$ |
| DHGR | $\mathbf{75.1 \pm 1.4}$ | $70.2 \pm 2.4$ | $81.7 \pm 1.4$ | $41.8 \pm 0.6$ | $37.6 \pm 0.5$ | $83.2 \pm 0.6$ |
| REFine | $74 \pm 2$ | $\mathbf{82.4 \pm 2.1}$ | $\mathbf{84.9 \pm 1.3}$ | $\mathbf{43.5 \pm 1.8}$ | $\mathbf{38.8 \pm 0.5}$ | $\mathbf{85.9 \pm 1.3}$ |
| APPNP | | | | | | |
| None | $49.4 \pm 1.7$ | $61.9 \pm 2$ | $62.1 \pm 1.3$ | $40.2 \pm 1.1$ | $35.4 \pm 0.7$ | $95.7 \pm 0.3$ |
| SDRF | $63.7 \pm 2.1$ | $77 \pm 1.4$ | $75 \pm 0.9$ | $41 \pm 1.1$ | $35.6 \pm 0.7$ | $95.8 \pm 0.2$ |
| FoSR | $55.1 \pm 1.5$ | $67 \pm 1.4$ | $68.4 \pm 1.8$ | $41.8 \pm 1$ | $35.7 \pm 0.6$ | $\mathbf{95.9 \pm 0.2}$ |
| BORF | $55.1 \pm 2$ | $65.1 \pm 2.4$ | $66 \pm 1.6$ | $39.6 \pm 0.8$ | $36.2 \pm 0.4$ | $95.5 \pm 0.2$ |
| DHGR | $70.8 \pm 1.6$ | $74.3 \pm 2$ | $81.7 \pm 2.2$ | $43 \pm 1$ | $37.9 \pm 0.5$ | $95.4 \pm 0.04$ |
| REFine | $\mathbf{74.6 \pm 1.5}$ | $\mathbf{82.4 \pm 1.7}$ | $\mathbf{86 \pm 1.4}$ | $\mathbf{44.5 \pm 1.2}$ | $\mathbf{38.8 \pm 0.8}$ | $95.7 \pm 0.2$ |

Table 7: Complete results with standard error of the mean (SEM) for datasets with more than 5,500 nodes. "T/O" indicates a timeout and "OOM" indicates out of memory. Best results are bolded.

|  | Actor | BGP | Tolokers | Roman-empire | Questions | Genius |
|---|---|---|---|---|---|---|
| Nodes | 7600 | 10k | 11k | 22k | 48k | 421k |
| $h$ | 0.21 | 0.28 | 0.59 | 0.04 | 0.84 | 0.59 |
| GCN | | | | | | |
| None | $28.4 \pm 0.2$ | $53.4 \pm 0.5$ | $77.2 \pm 0.3$ | $37 \pm 0.3$ | $65.7 \pm 0.4$ | $83.1 \pm 0.07$ |
| SDRF | $29.2 \pm 0.3$ | $53.9 \pm 0.5$ | $77.6 \pm 0.4$ | $46.2 \pm 0.2$ | OOM | OOM |
| FoSR | $28.1 \pm 0.2$ | $53.3 \pm 0.4$ | $77.4 \pm 0.3$ | $36.9 \pm 0.4$ | $63.3 \pm 0.3$ | $82.2 \pm 0.05$ |
| BORF | $28.3 \pm 0.3$ | $52 \pm 0.8$ | $77 \pm 0.3$ | $35.2 \pm 0.3$ | $65.9 \pm 0.5$ | T/O |
| DHGR | $\mathbf{31.4 \pm 0.3}$ | $57.3 \pm 0.4$ | $77.2 \pm 0.3$ | $56.1 \pm 0.2$ | $66.9 \pm 1.4$ | OOM |
| REFine | $31.3 \pm 0.4$ | $\mathbf{59.3 \pm 0.2}$ | $\mathbf{78 \pm 0.2}$ | $\mathbf{58.8 \pm 0.2}$ | $\mathbf{70.3 \pm 0.4}$ | $\mathbf{83.8 \pm 0.04}$ |
| GATv2 | | | | | | |
| None | $29.6 \pm 0.4$ | $62.3 \pm 0.3$ | $79.3 \pm 0.2$ | $14.8 \pm 0.4$ | $67.4 \pm 0.5$ | $81.7 \pm 0.1$ |
| SDRF | $29.7 \pm 0.3$ | $63.2 \pm 0.2$ | $\mathbf{79.9 \pm 0.2}$ | $20.8 \pm 0.2$ | OOM | OOM |
| FoSR | $29.2 \pm 0.5$ | $62.8 \pm 0.4$ | $79.5 \pm 0.3$ | $14.7 \pm 0.4$ | $\mathbf{67.6 \pm 0.5}$ | $81.2 \pm 0.06$ |
| BORF | $28.6 \pm 0.5$ | $63 \pm 0.3$ | $79.4 \pm 0.2$ | $14.9 \pm 0.4$ | $\mathbf{67.6 \pm 0.4}$ | T/O |
| DHGR | $32.8 \pm 0.3$ | $63.2 \pm 0.4$ | $79.2 \pm 0.2$ | $27.5 \pm 0.8$ | OOM | OOM |
| REFine | $\mathbf{35.1 \pm 0.3}$ | $\mathbf{63.3 \pm 0.3}$ | $79.7 \pm 0.3$ | $\mathbf{28.5 \pm 0.7}$ | $66.6 \pm 0.5$ | $\mathbf{83.6 \pm 0.03}$ |
| APPNP | | | | | | |
| None | $33.8 \pm 0.2$ | $63.6 \pm 0.5$ | $71.1 \pm 0.3$ | $14 \pm 0.07$ | $44.1 \pm 3.7$ | $81.9 \pm 0.4$ |
| SDRF | $33.8 \pm 0.2$ | $63.6 \pm 0.3$ | $71.8 \pm 0.2$ | $22.6 \pm 0.06$ | OOM | OOM |
| FoSR | $33.9 \pm 0.2$ | $63.6 \pm 0.5$ | $71.9 \pm 0.3$ | $14.3 \pm 0.4$ | $44.8 \pm 3.5$ | $81.2 \pm 0.4$ |
| BORF | $33.6 \pm 0.3$ | $63.4 \pm 0.3$ | $71.4 \pm 0.2$ | $15.5 \pm 0.6$ | $44.5 \pm 3.2$ | T/O |
| DHGR | $34 \pm 0.2$ | $63.8 \pm 0.4$ | $72.6 \pm 0.3$ | $28.3 \pm 0.6$ | OOM | OOM |
| REFine | $\mathbf{34.8 \pm 0.3}$ | $\mathbf{64.3 \pm 0.3}$ | $\mathbf{73.8 \pm 0.2}$ | $\mathbf{30.8 \pm 0.5}$ | $\mathbf{47 \pm 2.6}$ | $\mathbf{83.6 \pm 0.04}$ |

Table 8: Complete GCN results with standard error of the mean (SEM) for high-homophily datasets. Best results are bolded.

|  | Cora | Citeseer | Pubmed |
|---|---|---|---|
| Nodes | 2708 | 3327 | 19K |
| $h$ | 0.8 | 0.73 | 0.8 |
| None | $\mathbf{87.6 \pm 0.5}$ | $\mathbf{77.3 \pm 0.3}$ | $88.2 \pm 0.2$ |
| SDRF | $\mathbf{87.6 \pm 0.5}$ | $77.2 \pm 0.5$ | $\mathbf{88.3 \pm 0.2}$ |
| FoSR | $87 \pm 0.6$ | $76.7 \pm 0.4$ | $88.2 \pm 0.2$ |
| BORF | $86.6 \pm 0.6$ | $76.6 \pm 0.4$ | $87.3 \pm 0.2$ |
| REFine | $87.4 \pm 0.4$ | $77.2 \pm 0.4$ | $\mathbf{88.3 \pm 0.2}$ |

Table 9: Complete results with standard error of the mean (SEM) on heterophilic graphs for specialized GNNs vs. ST+REFine. "OOM" indicates out-of-memory. The best results are bolded, and the second-best are underlined.

|  | Cornell | Texas | Wisconsin | Chameleon | Squirrel | BlogCatalog | Actor | BGP |
|---|---|---|---|---|---|---|---|---|
| MixHop | $71.9 \pm 1.5$ | $79.1 \pm 1.7$ | $83.1 \pm 1.7$ | $43.2 \pm 1.3$ | $39.8 \pm 0.7$ | OOM | $\mathbf{36.2 \pm 0.3}$ | $64.3 \pm 0.2$ |
| H$_2$GCN | $73.2 \pm 1.8$ | $\mathbf{82.7 \pm 2}$ | $82.3 \pm 1.6$ | $\underline{41.8 \pm 1}$ | $40.4 \pm 0.4$ | $\mathbf{96.4 \pm 0.1}$ | $30.3 \pm 0.5$ | $64.9 \pm 0.5$ |
| GPRGNN | $\underline{70.8 \pm 1.2}$ | $81 \pm 1.7$ | $82.5 \pm 1$ | $40.9 \pm 1$ | $38.5 \pm 0.8$ | $95.7 \pm 0.1$ | $35.4 \pm 0.3$ | $\mathbf{65 \pm 0.5}$ |
| OrderedGNN | $70.8 \pm 2.1$ | $77.8 \pm 1.4$ | $82.1 \pm 1.1$ | $38 \pm 1.1$ | $34.3 \pm 0.7$ | $\underline{95.7 \pm 0.2}$ | $35.8 \pm 0.3$ | $\mathbf{65 \pm 0.1}$ |
| ST+REFine | $\mathbf{74.6 \pm 1.5}$ | $\underline{82.4 \pm 1.7}$ | $\mathbf{86 \pm 1.4}$ | $\mathbf{44.5 \pm 1.2}$ | $\mathbf{41.1 \pm 0.7}$ | $\underline{95.7 \pm 0.2}$ | $35.1 \pm 0.3$ | $64.3 \pm 0.3$ |

## H.2 Impact of Rewiring on Edge Homophily

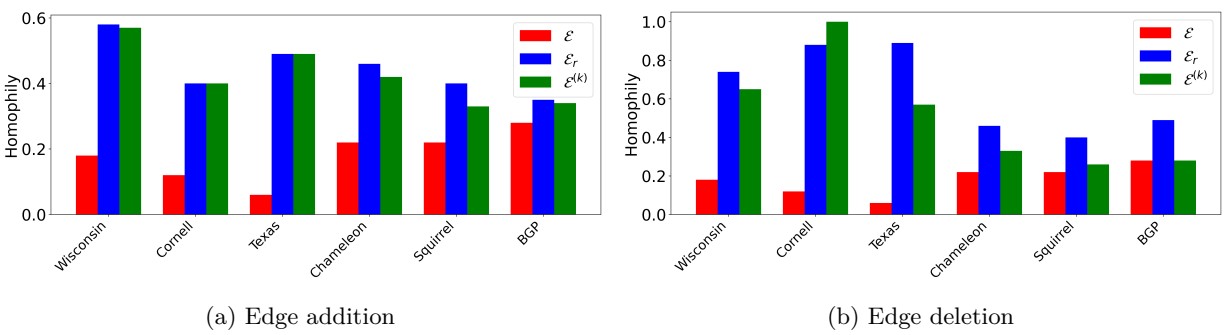

(a) Edge addition                    (b) Edge deletion

Figure 12: Edge homophily of the original edge set $\mathcal{E}$, the reference edge set $\mathcal{E}_r$ used for rewiring, and the rewired edge set $\mathcal{E}^{(k)}$.

## H.3 Reference Edge-Set Homophily: Features vs. Label-Driven Diffusion

Table 10 compares the homophily of the reference edge set $h(\mathcal{E}_r)$ when constructed using only node features ($\mathbf{\Gamma = D}$), using label-driven diffusion that incorporates both features and training labels ($\mathbf{\Gamma = PDP}$), and the baseline homophily of the original edge set $h(\mathcal{E})$. As shown, both reference edge sets exhibit consistently higher homophily than the original edge set. Moreover, incorporating label information ($\mathbf{\Gamma = PDP}$) further improves homophily in most cases compared to using features alone ($\mathbf{\Gamma = D}$).

Table 10: Homophily of the reference edge set $\mathcal{E}_r$ constructed using only node features ($\boldsymbol{\Gamma} = \mathbf{D}$), using label-driven diffusion ($\boldsymbol{\Gamma} = \mathbf{PDP}$), and the original edge set $\mathcal{E}$.

| | Cornell | Texas | Wisconsin | Chameleon | Squirrel | BlogCatalog | Actor | BGP | Roman-empire |
|---|---|---|---|---|---|---|---|---|---|
| $h(\mathcal{E})$ | 0.12 | 0.06 | 0.17 | 0.23 | 0.2 | 0.4 | 0.21 | 0.28 | 0.04 |
| $h(\mathcal{E}_r)$ ($\mathbf{D}$) | 0.47 | 0.55 | 0.58 | **0.28** | **0.29** | 0.4 | 0.24 | 0.4 | **0.52** |
| $h(\mathcal{E}_r)$ ($\mathbf{PDP}$) | **0.66** | **0.7** | **0.7** | 0.25 | 0.25 | **0.65** | **0.27** | **0.6** | 0.44 |

## I    Complexity and Runtime

The computation of the affinity kernels $\mathbf{D}$ and $\mathbf{P}$ has a time complexity of $O(d \cdot c^2)$, where $d$ is the dimension of the node feature vectors and $c$ is the cluster size, which we set to 100 or 500 in our experiments. The multiplication of the kernels to obtain $\boldsymbol{\Gamma}$ incurs a complexity of $O(c^3)$. The total number of clusters is given by $\frac{n}{c}$, where $n$ denotes the total number of nodes in the graph. Since the computational complexity per cluster is $O(c^3)$, the overall complexity is $O(c^3 \cdot \frac{n}{c}) = O(c^2 \cdot n)$, neglecting clustering via METIS (average-case $O(|\mathcal{E}|)$), which is negligible on standard sparse benchmarks ($|\mathcal{E}| = O(n)$). Notably, when $n \gg c$, the complexity simplifies to $O(n)$, indicating that the method remains linear in the number of nodes.

### I.1    Parallel Implementation on GPU

The bottleneck in our method is matrix multiplication. Since METIS partitions the graph into clusters of approximately equal size, it enables us to implement the matrix multiplication in parallel on the GPU. Thus, given $g$ units of GPU, the overall complexity is $O(c^3 \cdot \frac{n}{c} \cdot \frac{1}{g}) = O(\frac{c^2}{g} \cdot n)$, making our method even more efficient in practice compared to other approaches.

### I.2    Complexity Comparisons

Table 11: Comparison of method complexities: $n$ nodes, $c$ cluster size, $m$ edges, and $d_{\max}$ max node degree.

| Method | Complexity |
|---|---|
| SDRF | $O(md_{\max}^2)$ per edge |
| BORF | $O(md_{\max}^3)$ per cluster |
| FoSR | $O(n^2)$ per edge |
| REFine (Ours) | $O(c^3)$ per cluster |

### I.3    Runtime Comparisons

In Table 12, we compare the runtimes of our REFine method with baseline approaches across three datasets. Due to the high computational cost of the baselines on large datasets, we adapt them to use the same clustering strategy as REFine (described in Section 3.2). For small datasets (with fewer than 1000 nodes), we use the original implementations. For larger datasets (with 1000 or more nodes), we apply our clustering-based adaptations, using a cluster size of 500 for BlogCatalog and Roman-empire. For the runtime comparison, we use default hyperparameters for all methods. Specifically, for SDRF, we set $C^+ = 0.5$, $\tau = 100$, and the maximum number of iterations to $0.2n$, where $n$ is the number of nodes in the dataset (or the cluster size when applying the clustering adaptation). For FoSR, the maximum number of iterations is 50 for each cluster. For BORF, we set $h = 30$, $k = 20$, and $n = 3$.

Table 12: Runtime comparison across different methods. The table shows the runtimes (in seconds) for SDRF, FoSR, BORF, and our REFine method on three datasets: Wisconsin, BlogCatalog, and Roman-empire. $n$ represents the number of nodes and $m$ represents the number of edges. The smallest runtime for each dataset is highlighted in bold.

| Dataset | | | SDRF | FoSR | BORF | REFine (ours) |
|---|---|---|---|---|---|---|
| Name | $n$ | $m$ | | | | |
| Wisconsin | 251 | 900 | 0.8 | 4.7 | 3.8 | **0.1** |
| BlogCatalog | 5196 | $343k$ | 29 | 5.7 | 180 | **3.3** |
| Roman-empire | $22k$ | $65k$ | 20 | 5.8 | 380 | **4.2** |

### I.4 Negligible Overhead of METIS Clustering

REFine has per-cluster complexity $\mathcal{O}(c^3)$, and with $n/c$ clusters this yields $\mathcal{O}(c^2 n)$, where $n$ is the number of nodes and $c$ is the cluster size. Including clustering with METIS (average-case $\mathcal{O}(|\mathcal{E}|)$), the end-to-end complexity is $\mathcal{O}(|\mathcal{E}| + c^2 n)$. On standard sparse benchmarks ($|\mathcal{E}| = \mathcal{O}(n)$), when $c^2 n \gg |\mathcal{E}|$ the complexity reduces to $\mathcal{O}(c^2 n)$ and the $|\mathcal{E}|$ term is negligible.

For instance, in the Genius dataset ($n = 421k$, $|\mathcal{E}| = 989k$, $c = 100$), $c^2 n \gg |\mathcal{E}|$, and METIS runtime is negligible. The table below reports runtime (in seconds) for METIS, REFine, and competing methods using the same clustering, showing that METIS adds negligible overhead:

Table 13: Runtime (in seconds) of METIS clustering, REFine, and competing methods. Results on BGP and Roman-Empire show that METIS adds negligible overhead compared to the total runtime.

| Dataset | nodes | edges | METIS (clustering) | SDRF | FoSR | BORF | REFine (ours) |
|---|---|---|---|---|---|---|---|
| BGP | 10k | 206k | 0.09 | 84 | 5.79 | 95 | 1.75 |
| Roman-empire | 22k | 65k | 0.03 | 20 | 5.8 | 380 | 4.2 |

# J  Ablation Studies & Analysis

## J.1  Homophily and Rewiring Effectiveness

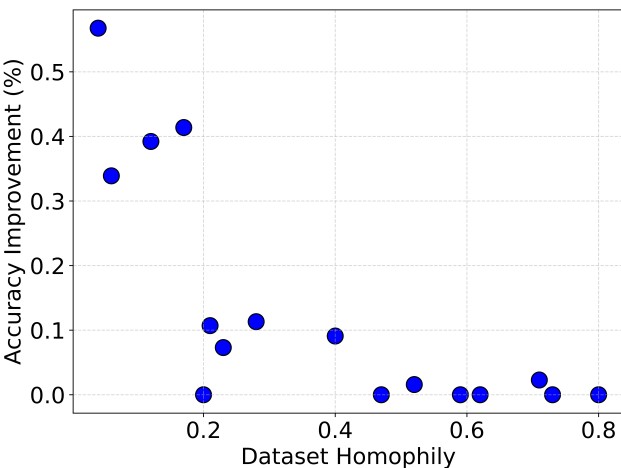

Figure 13: Test accuracy improvement across all evaluated datasets.

## J.2  Rewiring Only Using Train Labels

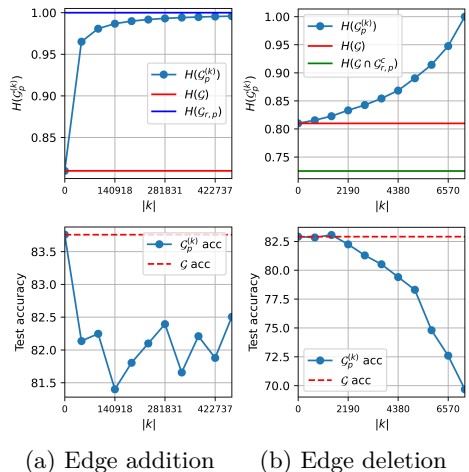

(a) Edge addition    (b) Edge deletion

Figure 14: Rewiring Cora using $\mathcal{E}_r$ built only from train labels.

## J.3  Cluster Size

In our experiments, we set the default cluster size to 500 for datasets with $1000 < n \leq 25000$ and 100 for datasets with $n > 25000$, primarily to optimize runtime. Table 14 presents the effect of varying the cluster size $\{100, 500, 1000, 2000\}$ when applying REFine. The "Improvement" column indicates whether changing the default cluster size leads to a statistically significant improvement. As shown, while increasing the cluster size beyond 500 can sometimes yield improvements, the overall effect remains relatively minor.

Table 14: Effect of varying the cluster size on REFine performance across different datasets. We report the mean test accuracy with the standard error of the mean (SEM) for different architectures (GCN, GATv2, APPNP). The "Improvement" column indicates whether increasing the cluster size results in a statistically significant improvement (V) or not (X).

| Dataset | $n$ | $h$ | Arch | 100 | 500 | 1000 | 2000 | Improvement |
|---------|-----|-----|------|-----|-----|------|------|-------------|
| Squirrel | 2223 | 0.2 | GCN | $40.5 \pm 0.8$ | $41.1 \pm 0.7$ | $40.7 \pm 0.6$ | N/A | X |
| | | | GATv2 | $37.4 \pm 0.6$ | $38.8 \pm 0.5$ | $38.9 \pm 0.5$ | N/A | X |
| | | | APPNP | $37.2 \pm 0.6$ | $38.8 \pm 0.8$ | $38.8 \pm 0.7$ | N/A | X |
| BlogCatalog | 5196 | 0.4 | GCN | $80.9 \pm 0.5$ | $85.2 \pm 0.3$ | $86.2 \pm 0.5$ | $86.4 \pm 0.3$ | V |
| | | | GATv2 | $80.7 \pm 0.7$ | $85.9 \pm 1.3$ | $83.1 \pm 1.2$ | $82.4 \pm 2.1$ | X |
| | | | APPNP | $95.9 \pm 0.3$ | $95.7 \pm 0.2$ | $95.3 \pm 0.2$ | $94.8 \pm 0.2$ | X |
| Roman-empire | 22k | 0.04 | GCN | $57 \pm 0.2$ | $58.8 \pm 0.2$ | $59.5 \pm 0.2$ | $59.7 \pm 0.2$ | V |
| | | | GATv2 | $27.4 \pm 0.8$ | $28.5 \pm 0.7$ | $28.5 \pm 1.1$ | $29.7 \pm 2$ | X |
| | | | APPNP | $29.7 \pm 0.7$ | $30.8 \pm 0.5$ | $29.2 \pm 0.8$ | $30.7 \pm 0.5$ | X |

## J.4 Sensitivity to the Gaussian-Kernel Scale $\epsilon$

We evaluate the sensitivity of REFine to $\epsilon$ using GCN on Wisconsin, BlogCatalog, and Actor, which span different graph sizes and selected kernel scales. For each fixed value of $\epsilon$, we select the remaining REFine hyperparameters on the validation set and report the resulting test accuracy.

Table 15: Sensitivity of REFine to the Gaussian-kernel scale $\epsilon$ using GCN. For each fixed $\epsilon$, the remaining REFine hyperparameters are selected on the validation set, and test accuracy is reported as mean $\pm$ SEM. Bold indicates the value used in the main experiments.

| Dataset | | GCN test accuracy | | | |
|---------|-----|-----|-----|-----|-----|
| Name | $n$ | $\epsilon = 10^{-1}$ | $\epsilon = 10^0$ | $\epsilon = 10^1$ | $\epsilon = 10^2$ |
| Wisconsin | 251 | $76.6 \pm 1.1$ | $78.4 \pm 1.1$ | $78.8 \pm 1.8$ | $\mathbf{82.5 \pm 1.6}$ |
| BlogCatalog | 5196 | $78.4 \pm 0.5$ | $81.3 \pm 0.6$ | $\mathbf{85.2 \pm 0.3}$ | $82.6 \pm 0.8$ |
| Actor | 7600 | $30.6 \pm 0.3$ | $\mathbf{31.3 \pm 0.4}$ | $30.0 \pm 0.4$ | $30.4 \pm 0.3$ |

The selected value performs best in all three cases, while values within one order of magnitude generally retain similar performance. This indicates that REFine is not highly sensitive to the exact value of $\epsilon$ once the appropriate scale has been identified.

