# OpenReview forum: "From Linking Homophily and Label Informativeness to Rewiring in GNNs"
_TMLR — Under review for TMLR_

### Review · Reviewer_rHmx · 2026-06-11

**Summary Of Contributions:**

- The paper formalizes edge homophily $h$, baseline homophily $h_0$, and label informativeness $LI$ through the same endpoint-label distribution, and proves a general lower bound relating $LI$ and $|h-h_0|$, which is a clean and useful framing
- It argues that increasing homophily can help message passing beyond LI alone, via a separability perspective and controlled rewiring simulations.
- It proposes a reference-edge-set rewiring method using features and training labels, together with conditional homophily-improvement propositions.
- Empirically, the paper evaluates REFine across a broad range of heterophilic benchmarks and reports useful runtime and cluster-size analyses

**Audience:**

Yes

**Audience Explanation:**

Researchers working on graph rewiring and heterophily in GNNs would likely be interested in the paper

**Claims And Evidence:**

No

**Claims Explanation:**

The paper has a useful practical idea: build a homophilic reference edge set from features and training labels, then use it to guide scalable graph rewiring. However, the main problem is that the narrative overstates what the theory and experiments actually establish.

- The stronger claim that high LI is unlikely or incompatible with low homophily is not established in general by Theorem 2.1. The theorem gives a lower bound, which shows only that LI must be nonzero when $h \ne h_0$; it does not upper-bound LI at low h. Think of a four-class balanced graph with edges (undirected) only between 1<->2 and 3<->4: here h=0 because no same-class edges exist, yet LI=1 because a neighbor label deterministically identifies the node label. This shows the paper's main narrative is false.
- The empirical evidence supports a weaker claim than the paper currently makes. REFine often improves standard GNNs and looks practically promising, but the text overstates superiority over both rewiring baselines and specialized heterophily GNNs. The tables contain multiple losses or non-wins, so "consistently outperforms" is not true.

**Requested Changes:**

1. Rewrite the main theory narrative so that it does not claim a general incompatibility between low homophily and high LI.
2. It would be better to add structured counterexamples or boundary-case experiments, including high-LI/low-h graphs such as deterministic heterophilic block pairs, and discuss them explicitly in the theory section.
3. Tone down the empirical claims and narrow "consistently outperforms" style language.

---

> ### Author Response · Authors · 2026-07-22
> **Response to Reviewer rHmx**
>
> Thank you for your helpful feedback. We appreciate the time you took to review our work and provide valuable suggestions.
> ## Scope of the Homophily - LI Claim
> We thank the reviewer for identifying this important issue. We agree that Theorem 2.1 does not establish a general incompatibility between low homophily and high label informativeness. It provides only a lower bound on LI in terms of $|h-h_0|$ and does not upper-bound LI when $h$ is low. Therefore, the reviewer’s deterministic heterophilic class-pair example is valid: a graph can have $h=0$ and $\mathrm{LI}=1$ when the neighbor label deterministically identifies the node label.
>
> Our intended claim instead applies under the balanced symmetric model of Theorem 2.2. Under this model,
> $$\mathrm{LI}(0)=\frac{\log\left(K/(K-1)\right)}{\log K},$$
> which is strictly below one for $K>2$ and approaches zero as the number of classes increases, whereas $\mathrm{LI}(1)=1$. Thus, under these assumptions, high LI becomes increasingly unlikely at low homophily as $K$ grows. We revised the paper throughout to make this scope explicit, clearly distinguish the conditional result from the general case, and present Figure 1 as empirical evidence that the considered real-world benchmarks approximately follow this restricted relationship.
>
> We also added the requested synthetic boundary-case experiment. We construct a balanced four-class stochastic block model (SBM) that begins with deterministic heterophilic class pairings, where $h=0$ and $\mathrm{LI}=1$. We gradually replace this structure with label-independent random mixing and then transition from random mixing to deterministic same-class connectivity, ending at $h=1$ and $\mathrm{LI}=1$. The expected degree, node features, data splits, and model initialization are controlled across the transition, and a GCN is trained at each point over multiple random seeds.
>
> The experiment confirms the reviewer’s observation that maximal LI can occur at zero homophily in the general case. It also yields the expected U-shaped LI curve and an approximately U-shaped accuracy curve. Importantly, for comparable LI values, the higher-homophily branch consistently achieves better GCN accuracy, supporting our narrower claim that homophily can affect message passing beyond what is captured by LI alone. We include the main results in the revised theory section and provide the full construction and an additional figure in the appendix.
>
> ## Empirical Performance Claims
> We agree that phrases such as "consistently outperforms" were too strong because REFine does not achieve the best result in every evaluated setting. We revised the abstract, introduction, experiment discussion, contributions, and conclusion to use precise and appropriately qualified language.

---

### Review · Reviewer_uuKH · 2026-07-10

**Summary Of Contributions:**

This paper adresses the question of the links between (edge) homophily h, label informativeness (LI) and possible classification accuracy.

A general lower bound for LI (compared to label entropy and homophily) is derived in the general case. For a toy model of graph data, an exact relationship is derived beteen LI and h. Real data is compatible with the trend of the relationship found.

This first theoretical part motivates the introduction of a rewiring method, that is explictly targetting to increase homophilly. Experiments convincingly demonstrate the relevance of this new technique.

The new method relies for a large part on the recently published method by Mendelman & Talmon (2025) (ICLR 2025), i.e. a very recent advance.

**Additional Comments:**

I did not check the proofs in the appendix.

I did not check for plagiarism, but the recent-ness of Mendelman & Talmon (2025) makes it unlikely.

**Audience:**

Yes

**Audience Explanation:**

Yes, although the assumptions about the graph task should be made explicit and early, see my first request.

**Claims And Evidence:**

Yes

**Claims Explanation:**

Yes, but improvements in the presentation are needed. Some details are missing from the presentation, see my requests below.

**Requested Changes:**

Comments are more or less in order of decreasing importance, not along the text order.

**Crucial changes needed**

- Graphs are versatile and many people work on graphs/GNNs. Here there are a mumber of assumptions about the data, that does not readily apply to all graph data. Although very classic graphs dataset are considered, the community has grown a lot and assumptions should be made very clear, early in the paper (at the level of introduction or right after it I would say).
    - edges are feature-less (I guess, otherwise how would the reference edge-set attributes would be initialized)
    - train/test split is among nodes of the same graph (not between graphs). I guess there are still multiple graphs per dataset, but training is not repeated on each of them, but shared? (the mention of "without loss of generality" should be suppressed, because in many graphs tasks, $\bar{n}=0,n$ (the whole graph is in train, or in test), making the proposed method pointless).
    - nodes must have attributes (in the paper, called features --ok--,  or sometimes embedding --I disapprove of that wording, which implies the vector is *learned*, see below.--)
    - node classification (extension to regression is not obvious)

- Graph rewiring is not consensual, please discuss the limitations inherent to graph rewiring at some point of the introduction.


- REFine hyperparameter $\epsilon$: a very broad range of multiple values was considered. It is necessary to report a study of how the choice impacts the results, and report the values used for each dataset (is the search repeated for each k chosen?).
This is actually an important point, because for any new method there is usually a set of hyperparameters associated. If tuning them is crucial to performance, the claims associated to ease of use, low resource consumption, etc, can be much decreased. On the other hand, if the values to be chosen are always in the same range for most datasets, and small variations of these do not impact restults, then it is fine.
The strategy of how epsilon is selected (one for all datasets --unlikely--, or one for each dataset only, or one for each dataset and each k value) must be mentionned in the main text.

- Figure 2: is the accuracy only that of the test-set nodes, or a mix of all nodes accuracy ? Make it explicit, so interpretation of the result is possible (showing only the test set accuracy is preferable of course, in my opinion).

- Similar point for homophily: the paragraph "Labels-only ablation", showing that "*as |k| increases,* [train-set only] *homophily
improves as expected but test accuracy declines*" (I added the brackets) made me realize that the changes in homophily reported throughout the paper are relating to train+val+test nodes' homophily. 1) Make this explicit when you define homophily or at some point of the main text when considering its variations. 2) Suggestion: why not also consider the test-set homophily? Meaning, only counting edges that have at least one node from the test set, ignoring the train-train edges.

- Section E.2 in appendix should be promoted to the main text, in the related works section, or even more prominently in the intro or conclusion, as a large part of this work is deeply connected withthe work of (Mendelman & Talmon, 2025).

- sec E.3: maybe make explicit here, and also in the main text, that the (true) homophily shown throughout the paper, was computed using also the test set labels (all labels).

- It seems that the node *embeddings* Z mentionned around Eq 4 are **not learned**, or if they were, the paper remains silent about how this embedding was learned. Either explain that this learning is needed before building the reference edge set, which I believe is not the case, or, change the wording to node feature, or better, node attribute, if those pertain to the dataset.
Later in section 3.2 the term "node feature vectors" is used, this is more clear, but the previous use of *embedding* makes it confusing still.


**Minor modifications needed**

- recipe for building the matrices D, D1, D2: I appreciate that the text does not extend too much on this point, but having the maths written down in appendix would be nice, for clarity and completeness (with maths notations instead of words like "summing rows/columns").

- Figure 2: which GNN was used? Mention it in the caption.

- Suggestion: "entropy of the random endpoint label C, .." Here, maybe, take the time to introduce $C$ as a random variable (I initially thought it was a value, and was confused for 1 minute)

- Appendix C, figures 4a,b,c: it is not clear what is changed to produce various reference edge sets. When reading the main text, I was under the impression that there was no hyper-parameter other than $\epsilon$ to build $\varepsilon_{r}$. Here there is the additional parameter $p$, but the value used are not mentionned in the caption nor in the text.
Same thing for figs 6,7,8: please provide the values of $p$, at least in the captions.

- below corrolary 3.2, Edge deletion : mention that c denotes the complement (it is done, but much later in the paper)

**Scientific Suggestion (fully optional)**

- Suggestion: discuss the link (maybe simply empirically on a few datasets) between homophilly $h$ and the Dirichlet Energy of the label distribution, $Tr(Y \mathcal{L} Y)$.

---

> ### Author Response · Authors · 2026-07-22
> **Response to Reviewer uuKH (1/3)**
>
> We thank the reviewer for the constructive feedback and helpful suggestions, which have helped us improve the clarity and quality of the paper.
> ## Graph Setting and Scope
> We revised the opening paragraph of the Introduction to explicitly state that we consider transductive node classification with observed node features, featureless edges, and disjoint training, validation, and test subsets of nodes in the same graph. Each benchmark dataset considered here consists of a single graph, as is standard in commonly used transductive node-classification benchmarks; different splits partition the nodes of that graph, and a separate model is trained for each split. We also removed "without loss of generality" from Section 3.2 and now introduce the node indexing specifically within the setting considered in this work.
>
> ## Limitations of Graph Rewiring
> We thank the reviewer for this suggestion. We added a separate discussion in the Introduction explicitly acknowledging that graph rewiring remains a debated design choice and is not universally appropriate. We also outline its inherent limitations, including the risk of removing task-relevant relations, introducing spurious ones, or altering relations that should be preserved, and clarify that rewiring may therefore not improve downstream performance in every setting.
>
> ## REFine Hyperparameter $\epsilon$
> We thank the reviewer for raising this important point. Hyperparameter selection is standard in graph-rewiring methods, for example, SDRF also requires tuning several rewiring parameters. We agree, however, that our original presentation did not sufficiently explain how $\epsilon$ is selected or how its choice affects performance.
>
> For each dataset, we select $\epsilon$ on the validation set using a fixed value of $k$ at the center of the considered range, and then keep $\epsilon$ fixed while selecting $k$. Thus, $\epsilon$ is selected once per dataset and is not re-selected for each value of $k$. We now clarify this protocol and report the selected value for every dataset in the Experiment Settings appendix.
>
> Since $\epsilon$ controls the bandwidth of a Gaussian kernel, its scale can also be estimated using the common median-distance heuristic, which sets $\epsilon$ to the median squared pairwise distance between node features. We now report the corresponding median-based scale for every dataset:
> | Dataset     | $\epsilon$ used | $\epsilon_{\mathrm{med}}$ |
> |-------------|:---------------:|:-------------------------:|
> | Cornell     |     $10^{2}$    |     $1.28\times10^{2}$    |
> | Texas       |     $10^{2}$    |     $1.04\times10^{2}$    |
> | Wisconsin   |     $10^{2}$    |     $1.25\times10^{2}$    |
> | Chameleon   |     $10^{2}$    |     $1.70\times10^{1}$    |
> | Squirrel    |     $10^{1}$    |     $2.40\times10^{1}$    |
> | BlogCatalog |     $10^{1}$    |     $1.11\times10^{2}$    |
> | Actor       |     $10^{0}$    |     $8.00\times10^{0}$    |
> | BGP         |     $10^{1}$    |     $1.09\times10^{1}$    |
> | Tolokers    |    $10^{-2}$    |    $1.31\times10^{-1}$    |
> | Questions   |    $10^{-5}$    |    $3.91\times10^{-6}$    |
> | Genius      |    $10^{-8}$    |    $2.39\times10^{-4}$    |
>
> For $10$ of the $11$ datasets, the selected and median-based values lie in the same or adjacent orders of magnitude, indicating that this heuristic can generally restrict the search to a few nearby scales rather than the full numerical range.
>
> As requested, we also added a sensitivity ablation to the Ablation Studies appendix. For each fixed value of $\epsilon$, we select the remaining REFine hyperparameters on the validation set and report test performance. The selected value performs best in all cases, while values within one order of magnitude generally retain comparable performance. This indicates that REFine is not highly sensitive to the exact value of $\epsilon$ once the appropriate scale has been identified.
>
> ## Accuracy in the Cornell Homophily--LI Experiment
> We thank the reviewer for requesting this clarification. In the figure showing homophily, LI, the LI lower bound, and GNN accuracy on Cornell as a function of the number of added same-class edges, the accuracy is computed exclusively over the held-out test-set nodes and does not include training or validation nodes. We revised the figure caption to state this explicitly.

---

> ### Author Response · Authors · 2026-07-22
> **Response to Reviewer uuKH (2/3)**
>
> ## Reported and Estimated Homophily
> We thank the reviewer for pointing out this ambiguity. Throughout the paper, the reported homophily is the full-graph edge homophily, computed post hoc over the complete edge set using the ground-truth labels of all nodes, including validation and test nodes. This full-label quantity is used only for evaluation and is not available to the rewiring procedure. We now make this explicit both when defining homophily in the main text and in Appendix F.4.
>
> Appendix F.4 considers a different quantity: an estimate of the true full-graph homophily constructed using training and validation labels, without access to test labels. We revised this subsection to clearly distinguish the homophily estimate available for checking the rewiring conditions from the true homophily reported post hoc in our empirical analyses.
>
> We agree that homophily restricted to edges incident to test nodes could provide a complementary, split-specific diagnostic. Throughout the paper, we use full-graph edge homophily when reporting or analyzing structural changes in the graph, including the homophily curves and the evaluation of the rewiring conditions. Predictive performance, including in the labels-only ablation, is evaluated separately using accuracy on the held-out test nodes. We use this distinction because full-graph homophily is the structural quantity appearing in our definitions and theoretical guarantees, while held-out test performance measures generalization. The labels-only ablation further illustrates that improving full-graph homophily does not necessarily improve test performance.
>
> ## Connection to Label-Driven Diffusion
> We moved the discussion from Appendix E.2 to the Proposed Method subsection. We now clarify the relationship between our construction and label-driven diffusion, as well as the design choices specific to graph rewiring.
>
> ## Role of $\mathbf{Z}$ in Eq. (4)
> We thank the reviewer for pointing out this ambiguity. No node embedding is learned before constructing the reference edge set. The matrix $\mathbf{Z}$ in Eq. (4) is a generic node-representation matrix used only in the theoretical analysis, it may represent hidden or output representations produced by a GNN, but the theorem does not assume any particular architecture or learning procedure.
>
> The theorem is a conditional statement: if $\mathbf{Z}$ is linearly separable, then its Dirichlet energy is lower-bounded by a quantity proportional to $1-h$. Thus, at lower homophily, any linearly separable representation must vary more strongly across graph edges. Since message passing tends to smooth node representations, this formalizes the tension between smoothness and class separability on heterophilic graphs and explains why increasing homophily can make these objectives more compatible.
>
> The theorem provides motivation for homophily-enhancing rewiring but is not a step of REFine: the reference edge set is constructed directly from the observed node features $\mathbf{X}$ and the available training labels. We revised the text around Eq. (4) to replace the potentially confusing term "learned embeddings" with "node representations" and to make this distinction explicit.

---

> ### Author Response · Authors · 2026-07-22
> **Response to Reviewer uuKH (3/3)**
>
> ## Minor Clarity and Completeness Revisions
> We thank the reviewer for these helpful suggestions. We added explicit mathematical definitions of $\mathbf{D}_1$, $\widetilde{\mathbf{D}}$, $\mathbf{D}_2$, and $\mathbf{D}$ to the Additional Method Details appendix. We also revised the caption of Figure 3 to specify that the reported node-classification accuracy is obtained using a GCN. To make the notation easier to follow, we added local reminders that $C$ is the random endpoint-label variable and that $\mathcal{E}_r^c$ denotes the complement of $\mathcal{E}_r$, with both definitions referring back to Subsection 2.1.
>
> We further clarified the role of $p$ in the controlled simulations. These simulations begin with the ideal same-label reference edge set, which contains all same-class node pairs and no different-class pairs. To generate $\mathcal{E}_{r,p}$, this ideal set is randomly corrupted according to $p$: same-class pairs may be removed and different-class pairs may be added. Thus, $p$ controls the amount of corruption applied to the ideal reference edge set, producing reference edge sets with lower homophily as $p$ increases. These edge sets are generated solely to validate the theoretical rewiring conditions over a controlled range of reference-set homophily values; they are not constructed by REFine, and $p$ is therefore not a REFine hyperparameter. We now state this distinction explicitly and report the values of $p$ in all relevant figure captions.
>
> ## Homophily and Label Dirichlet Energy
> We thank the reviewer for this particularly insightful suggestion. For an unweighted graph and the one-hot label matrix $\mathbf{Y}$, the two quantities are related exactly by
> $$\frac{\operatorname{tr}(\mathbf{Y}^{\top}\mathcal{L}\mathbf{Y})}{\sum_{u,v}A_{u,v}}=1-h.$$
>
> Thus, higher homophily corresponds exactly to a smoother label signal over the graph, with less label variation across edges.
>
> This provides a direct smoothness interpretation of homophily and complements Theorem 2.3, which connects homophily to the Dirichlet energy required for linearly separable node representations. We added the derivation and its interpretation to  Appendix A.3.

---

### Review · Reviewer_kH5P · 2026-07-14

**Summary Of Contributions:**

This paper studies graph rewiring from the perspective of homophily. The authors first investigate the relationship between edge homophily and label informativeness (LI) through the joint distribution of endpoint labels. Then they establish a lower bound relating homophily to the Dirichlet energy of node embeddings and finally propose a homophily-enhancing graph rewiring framework based on a diffusion kernel to improve message passing.

I have several concerns regarding the theoretical contributions and how they support the overall claims.

**1. The connection between the theoretical components is not well established**. The paper motivates its theoretical analysis by asking which graph properties make message passing effective, but the central theorem concerns only the joint distribution of node labels. Since message passing operates on node features and hidden representations, the connection between the theorem and the claimed implications for GNN performance remains largely intuitive rather than formally established. More broadly, the theoretical narrative proceeds from "relating homophily to label informativeness", to "bounding the Dirichlet energy using homophily", and finally to "concluding improved message passing". However, these components are analyzed separately, and the paper does not provide a convincing channel that formally links them into a unified argument. As a result, the overall theoretical justification is weaker than the paper's presentation suggests.

**2. The conceptual novelty of relating homophily and label informativeness is somewhat limited**. Both edge homophily and label informativeness are functionals of the same joint distribution of endpoint labels. Homophily measures the probability mass on the diagonal, while label informativeness measures the deviation of the joint distribution from independence. Consequently, their relationship appears natural (the precise inequality may be not). The qualitative connection (Theorem 2.1) itself is expected from the definitions and follows from a standard information-theoretic argument. Moreover, this analysis is entirely label-based and does not involve node features or message passing, making its connection to the subsequent GNN analysis irrelevant.

**3. The rewiring framework is conditional on the existence of a good reference graph**. The theoretical guarantees assume that the constructed reference edge set already has higher homophily than the original graph. However, the key algorithmic contribution of the paper is precisely the construction of this reference graph via the proposed diffusion kernel. In other words, the difficult part is not proving that rewiring improves homophily once such a reference graph is available but rather explaining why the proposed construction produces a reference graph with the desired properties. This crucial step is justified primarily by intuition and empirical evidence, leaving a significant gap between the algorithm and its accompanying theory.

**Audience:**

Yes

**Audience Explanation:**

Topics such as graph rewiring and graph neural networks under heterophily are of clear interest to the TMLR community.

**Broader Impact Concerns:**

N.A.

**Claims And Evidence:**

No

**Claims Explanation:**

See my concerns in Summary of Contributions.

**Requested Changes:**

1. I would encourage the authors to better articulate what genuinely new insight this inequality in Section 2 provides beyond the well-known observation that both statistics derive from the same joint distribution.

2. A much stronger theoretical contribution would analyze the actual message-passing operator. For example, one could study and derive how the expected embedding evolves under different homophily regimes. Alternatively, one could bound the Bayes classification error after one aggregation step as a function of the edge-label transition matrix.

---

> ### Author Response · Authors · 2026-07-22
> **Response to Reviewer kH5P (1/2)**
>
> We thank the reviewer for the careful feedback. While we respectfully disagree with some of the conclusions, the comments helped us clarify the scope, novelty, and role of the theoretical results. We revised the paper accordingly and address each concern below.
>
> ## Relationship Between the Theoretical Components
> We thank the reviewer for raising this concern. We respectfully clarify that the label-distribution results are not intended, by themselves, to prove improved GNN performance. Rather, the paper presents two complementary mechanisms through which homophily is relevant to message passing.
>
> The first mechanism concerns the neighborhood-label distribution. Theorems 2.1 and 2.2 characterize when homophily constrains the amount of label information carried by graph neighborhoods. Following prior work such as [1] and [2], we study these label-based structural properties because they characterize whether neighborhoods carry class-relevant information, independently of a particular choice of node features, hidden representations, or GNN architecture.
>
> The second mechanism is representation-level. Theorem 2.3 directly relates homophily to the minimum Dirichlet energy compatible with a linearly separable node representation. When homophily is lower, a linearly separable representation must retain greater variation across graph edges, making it harder to reconcile class separability with the smoothing tendency of message passing.
>
> These mechanisms are complementary rather than consecutive. The first explains when homophily also constrains LI, while the second explains why homophily can be beneficial even when LI does not improve. Figure 3 illustrates precisely this distinction, showing that, for $k<191$, GCN test accuracy increases with homophily while LI decreases. Combining the two results provides a broader picture of homophily than either alone: homophily affects both the class information carried by neighborhoods and the compatibility between representation smoothness and class separability. Specifically, after Theorem 2.3, we added an explicit summary clarifying that the LI analysis characterizes when homophily constrains the label information carried by graph neighborhoods, whereas the smoothness result characterizes how homophily affects the compatibility between the smoothing tendency of message passing and class separability.
>
> We appreciate the reviewer's suggestion to analyze a specific message-passing operator or its Bayes error. Such an analysis would provide a complementary, model-dependent perspective, but would address a different question from the architecture-independent graph-level and representation-level mechanisms studied here. We therefore leave this direction for future work.
>
> [1] Ma, Yao, et al. "Is homophily a necessity for graph neural networks?." arXiv preprint arXiv:2106.06134 (2021).
>
> [2] Platonov, Oleg, et al. "Characterizing graph datasets for node classification: Homophily-heterophily dichotomy and beyond." Advances in Neural Information Processing Systems 36 (2023): 523-548.

---

> ### Author Response · Authors · 2026-07-22
> **Response to Reviewer kH5P (2/2)**
>
> ## Novelty of the Homophily - LI Characterization
> We agree that the shared dependence of homophily and LI on the endpoint-label distribution $\mathbf{\Pi}$ makes it natural to ask whether the two quantities can be related. We respectfully disagree, however, that a useful relationship follows directly from this observation. Two scalar functionals of the same high-dimensional distribution need not determine or meaningfully constrain one another. Indeed, homophily does not determine LI in general: $h=0$ can coexist with $\mathrm{LI}=1$, and the same homophily value can correspond to substantially different LI values depending on how the probability mass is distributed across class pairs.
>
> Our contribution is therefore to formalize this coupling and characterize both what can and cannot be inferred. Theorem 2.1 establishes the general bound
> $$\mathrm{LI}\geq\frac{2}{\mathcal{H}(C)}(h-h_0)^2.$$
>
> identifying $h_0$ as the independence baseline and quantifying how the guaranteed minimum LI grows quadratically with the deviation $|h-h_0|$. We then identify a balanced symmetric model for which Theorem 2.2 gives an exact closed-form relationship between homophily and LI, including its dependence on the number of classes. Figure 1 further shows that many commonly used real-world benchmarks approximately follow this theoretical family.
>
> To the best of our knowledge, prior neighborhood-label-distribution and LI analyses, do not derive this general bound, the balanced-model closed form, or the empirical alignment with this family. Thus, while the proof uses standard information-theoretic tools, the formal relationship we establish between these graph quantities and the regimes we identify are new. Specifically, at the end of the homophily - LI analysis, we added a paragraph clarifying that our contribution goes beyond observing that homophily and LI are both computed from $\mathbf{\Pi}$: it includes the general quantitative bound in Theorem 2.1, the exact closed-form relationship under the balanced model we characterize in Theorem 2.2, and the empirical finding in Figure 1 that many real-world benchmarks approximately follow this theoretical family.
>
> ## Conditional Guarantees and the Reference Edge Set
> We agree with the reviewer that the rewiring guarantees are conditional on the quality of the reference edge set. This is precisely the purpose of the general framework: Propositions 3.1 and 3.3 characterize the sufficient conditions under which a given reference edge set improves homophily through edge addition or deletion. REFine then provides one practical construction of such a reference edge set using node features and the available training labels.
>
> Without assumptions connecting the observed features and available labels to the unknown labels, a construction based only on these inputs cannot be guaranteed to produce a highly homophilic reference edge set on arbitrary data. We therefore do not assume that the sufficient conditions always hold. Instead, Appendix F.4 provides a sampling procedure for estimating the relevant homophily quantities from the available training and validation labels and shows that these estimates closely approximate the corresponding post-hoc full-label values. This allows the conditions of the rewiring framework to be checked in practice rather than assumed.
>
> In addition, Appendix H.2 reports the homophily of the reference edge sets produced by our construction across real-world benchmarks and shows that the homophily condition required by the framework holds in many evaluated cases. Appendix C separately controls the quality of the reference edge set and validates the predicted behavior of edge addition and deletion. Thus, the proposed construction is supported by empirical verification rather than intuition. We revised the paper to distinguish more clearly between the general conditional framework, our practical reference-edge-set construction, and the procedure used to check whether the sufficient conditions hold.

---

### Comment · Reviewer_uuKH · 2026-07-15
**Discussion**

I very much agree with the other points by the other 2 referees and invite the authors to address all the points raised and come back to us then.